# Constrained Bayesian Experimental Design via Online Planning

**Yujia Guo** [1 2]  **Daolang Huang** [1 2]  **Xinyu Zhang** [1 2]  **Sammie Katt** [1 2]  **Samuel Kaski**[* 1 2 3]  **Ayush Bharti**[* 2]

## Abstract

Bayesian experimental design (BED) is a principled framework for data-efficient design of sequential experiments. However, existing BED methods are unable to adapt to dynamic constraints inherent in real-world tasks due to budget limitations, varying costs, or physical constraints that restrict how designs evolve over time. In this paper, we introduce a novel approach to BED that enables constrained optimization of experimental designs by combining *offline* pre-training of an amortized policy and a posterior network with *online* multi-step lookahead planning using scenario trees. We empirically demonstrate that our method yields substantially more informative design sequences than existing methods across a range of constrained BED tasks, while incurring only a modest additional computational overhead.

## 1. Introduction

Many scientific disciplines such as drug discovery (Lyu et al., 2019; Masood et al., 2024), material design (Frazier & Wang, 2015), and clinical trials (Cheng & Shen, 2005) rely on sequential experimentation to learn about complex underlying processes. *Bayesian experimental design* (BED) (Chaloner & Verdinelli, 1995; Rainforth et al., 2024) provides a principled model-based framework for adaptively selecting optimal experiments under uncertainty. The most commonly used information-theoretic utility in BED is the *expected information gain* (EIG) (Lindley, 1956; 1972), which quantifies the expected reduction in entropy of the posterior uncertainty over unknown quantities, or equivalently, the *mutual information* between the experimental observables and the quantities of interest (Ryan et al., 2016).

While theoretically appealing, estimating and optimizing

---

[*]Equal contribution [1]ELLIS Institute Finland [2]Department of Computer Science, Aalto University, Finland [3]Department of Computer Science, University of Manchester, UK. Correspondence to: Yujia Guo <yujia.guo@aalto.fi>.

*Proceedings of the 43rd International Conference on Machine Learning*, Seoul, South Korea. PMLR 306, 2026. Copyright 2026 by the author(s).

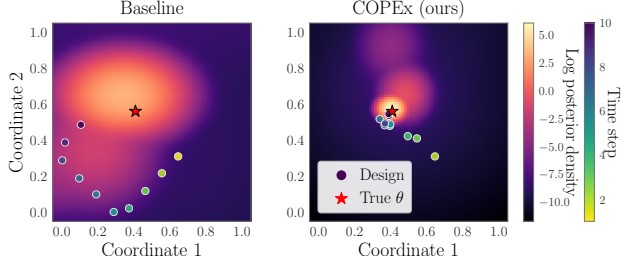

*Figure 1. BED under design constraints in location finding task.* When consecutive designs are forced to be at most a pre-defined distance apart (0.1 in this case, see Section 4.1), the baseline amortized policy of Huang et al. (2026) (*left*) inefficiently explores the design space, leading to high posterior uncertainty. In contrast, our constrained online planning method, named COPEx (*right*), yields more informative designs and better posterior concentration.

EIG is challenging as it involves nested expectations (Rainforth et al., 2018) and repeated posterior updates, which is especially burdensome in sequential settings. To address this issue, several BED methods optimize tractable lower bounds on EIG in a *myopic* manner (Foster et al., 2019; 2020). More recently, *amortized* BED methods (Foster et al., 2021; Ivanova et al., 2021; Blau et al., 2022; Lim et al., 2022; Blau et al., 2023; Iqbal et al., 2024; Shen et al., 2025; Bracher et al., 2026) train a design policy offline to maximize total EIG over the design sequence, enabling instantaneous proposal of non-myopic designs at test time.

Despite the methodological advancements, existing BED approaches overlook the fact that experimental design is often subject to constraints during deployment. Measurement costs can vary across instruments and laboratories, budgets may change over time, and operational constraints such as feasibility restrictions or limited movement ranges often dictate the set of admissible future experiments (Zhu et al., 2021; Popović et al., 2020). Crucially, these costs and constraints are not merely operational details but fundamentally reshape the optimal experimental strategy. This is exemplified in Figure 1(*left*) for the location finding task (see Section 4.1), where the method of Huang et al. (2026) performs sub-optimally under a transition constraint, i.e., when consecutive designs cannot change arbitrarily. Such constraints occur in mobile sensing applications (Atanasov et al., 2014) wherein repositioning a sensor incurs travel time and energy, and so a smooth trajectory is preferred.

Adapting to dynamic costs, budget limits, and feasibility constraints poses a unique challenge for BED. First, the problem can no longer be treated as a sequence of independent designs and requires a non-myopic solution as the utility of each experiment depends on its downstream effect. Second, each constraint configuration leads to a different optimal solution and, thus, would require rerunning the expensive offline computation process in amortized solutions.

In this paper, we introduce **C**onstrained **O**nline **P**lanning for **Ex**perimental design (COPEx), a semi-amortized method that solves the constrained BED problem via multi-step lookahead search over a scenario tree of hypothetical experimental outcomes (Høyland & Wallace, 2001). A naive implementation of such planning is computationally prohibitive, since evaluating candidate trajectories requires recursively estimating the posterior and its EIG at every node of the tree. We overcome this bottleneck by learning an amortized posterior (or posterior-predictive) network that enables fast posterior calculation, efficient generation of fantasy outcomes, and a tractable surrogate for EIG along simulated trajectories. Finally, to improve computational efficiency under limited planning budgets, COPEx uses an amortized design policy as a proposal distribution to bias search toward high-utility regions of the design space. Our results in Section 4 demonstrate that COPEx successfully adapts to the constraints and consistently achieves superior information gain compared to baselines in a variety of BED and active learning tasks with modest computational overhead.

## 2. Related Work

BED methods, such as by Rainforth et al. (2018); Foster et al. (2019; 2020); Kleinegesse & Gutmann (2019; 2020); Kleinegesse et al. (2021), that optimize approximations of EIG in a *myopic* manner are inherently incapable of reasoning about long-term feasibility under budgets or design-dependent constraints. Traditional policy-based sequential BED methods have leveraged dynamic programming principles for *non-myopic* planning. However, they either require substantial computational resources when maximizing EIG (Huan & Marzouk, 2016), or deal with simpler objectives (Carlin et al., 1998; Brockwell & Kadane, 2003; Murphy, 2003; Müller et al., 2007) and linear-Gaussian problems (Ben-Gal & Caramanis, 2002). Importantly, none of these methods handles design-dependent or budget constraints.

To address the computational burden of planning in non-myopic BED, policy-based *amortized* methods train neural design policies offline to maximize total EIG across experiment sequences, enabling non-myopic design selection with low test-time latency. This line of work was initiated by Foster et al. (2021) and extended to handle implicit likelihood models (Ivanova et al., 2021), non-differentiable simula-

tors (Lim et al., 2022), discrete design spaces (Blau et al., 2022), and downstream decision-focused objectives (Huang et al., 2024). Several subsequent approaches further combine amortized design selection with amortized inference components, for example, by learning the posterior or the posterior-predictive surrogates used within the design objective (Blau et al., 2023; Iollo et al., 2024; Shen et al., 2025; Huang et al., 2026; Bracher et al., 2026). More recently, Hedman et al. (2025) propose a semi-amortized variant that periodically refines the policy at test time.

Despite their non-myopic training objectives, these methods typically assume a *fixed* notion of feasibility, cost, and utility during training. When constraints or budgets change at deployment, the resulting policies generally cannot adapt without additional online optimization or retraining. For instance, continuous policy methods (Foster et al., 2021) require architectural redesigns or explicit regularization to be constraint-aware. Pool-based methods (e.g., Huang et al., 2026) can enforce hard constraints by masking invalid actions. However, this post-hoc restriction forces the policy onto trajectories unseen during training, leading to significantly sub-optimal designs as shown in Figure 1.

In contrast to BED, constraint-aware optimization has received substantial attention in Bayesian optimization (BO) and active learning (AL). Early approaches incorporate evaluation costs by modifying acquisition functions (Snoek et al., 2012), with subsequent refinements including cost-cooling strategies (Lee et al., 2020b) and multi-objective formulations (Abdolshah et al., 2019). Lee et al. (2021) extend non-myopic BO (Lee et al., 2020a; Jiang et al., 2020a) to a cost-constrained setting. Gardner et al. (2014); Lam & Willcox (2017) propose a lookahead approach for BO with inequality constraints, while Jiang et al. (2020b) propose a one-shot multi-step tree BO method that renders non-myopic lookahead tractable by jointly optimizing all decision variables on a scenario tree, avoiding nested dynamic programs. Building on this, Astudillo et al. (2021) propose a method with unknown, heterogeneous evaluation costs under hard budget constraints, and Xie et al. (2024) further connect cost-aware BO to Pandora's Box, yielding Bayes-optimal policies for discrete uncorrelated settings. Closely related to transition-constrained settings, Yang et al. (2024) propose an amortized BO policy that generates optimization trajectories under movement costs. However, extending these methods to sequential BED is non-trivial. Unlike BO, where utilities can often be computed from the Gaussian processes' predictive mean and variance with analytic posterior updates, estimating the EIG objective and the posterior entropy repeatedly across every node of the outcome-branching lookahead trees can be prohibitively expensive and unstable.

Another type of constraint studied extensively is safety. Safe

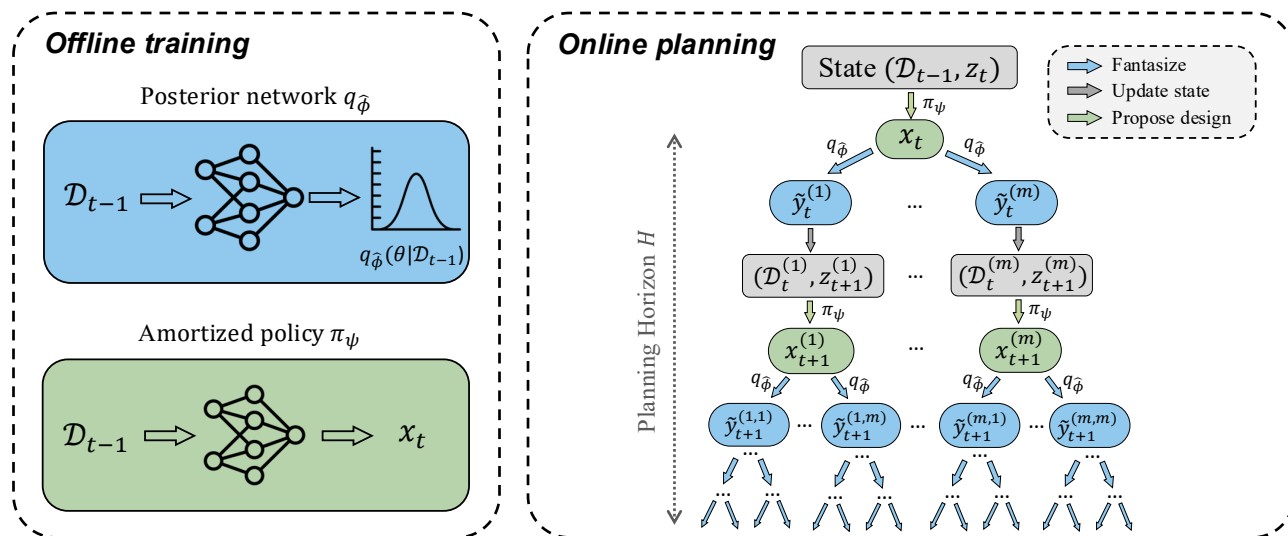

*Figure 2. Offline training and online planning for COPEx.* During *offline training*, we pre-train an amortized posterior network $q_{\hat\phi}$ to enable rapid state updates and sample future observations, alongside a design policy $\pi_{\hat\psi}$ that generates high-quality initial proposals. During *online planning*, COPEx leverages these amortized modules to efficiently construct a multi-step lookahead scenario tree.

BO methods (Sui et al., 2015; 2018; Berkenkamp et al., 2023) enforce high-probability safety constraints by modeling unknown constraint functions, while safe AL approaches incorporate analogous constraints into data acquisition for regression (Zimmer et al., 2018; Li et al., 2022; 2026). Separately, recent work in simulation-based Bayesian inference accounts for heterogeneous computational costs while sampling (Bharti et al., 2025), but does not address constrained design decisions. Overall, while cost- and safety-aware methods in BO or AL provide useful tools, principled methods for sequential BED that enforce dynamic constraints online through belief updates remain largely underexplored.

## 3. Method

We begin by formulating the constrained BED problem in Section 3.1, and then present our planning objective via multi-step scenario trees in Section 3.2. To enable online construction of the tree, we amortize posterior inference and use an offline-trained policy to initialize it (Section 3.3). Finally, Section 3.4 describes how we jointly optimize all decision variables over the scenario tree. An overview of the method is in Figure 2 and the algorithm is in Algorithm A1.

### 3.1. Problem Formulation

We consider sequential BED for a model $p(y\,|\,x,\theta)$ with parameters $\theta \in \Theta \subseteq \mathbb{R}^{d_\Theta}$, prior $p(\theta)$, designs $x \in \mathcal{X} \subseteq \mathbb{R}^{d_\mathcal{X}}$, and observations $y \in \mathcal{Y}$. At each step $t$, the agent selects $x_t$ and observes $y_t \sim p(\cdot\,|\,x_t,\theta)$. Let the complete history up to step $t$ be denoted as $\mathcal{D}_t = \{(x_i,y_i)\}_{i=1}^t$, with $\mathcal{D}_0 = \emptyset$. In the unconstrained setting, $x_t$ is chosen by maximizing

the EIG, i.e., $x_t^\star = \arg\max_{x_t \in \mathcal{X}} \text{EIG}(x_t; \mathcal{D}_{t-1})$, where:

$$\text{EIG}(x_t; \mathcal{D}_{t-1}) = \mathbb{E}_{p(y_t\,|\,x_t,\mathcal{D}_{t-1})}\big[\mathcal{H}[p(\theta\,|\,\mathcal{D}_{t-1})]$$
$$- \mathcal{H}[p(\theta\,|\,\mathcal{D}_{t-1} \cup \{(x_t,y_t)\})]\big]. \quad (1)$$

Here, $\mathcal{H}[\cdot]$ represents the entropy of a distribution, and the expectation is taken over the marginal predictive distribution

$$p(y_t|x_t,\mathcal{D}_{t-1}) = \mathbb{E}_{p(\theta\,|\,\mathcal{D}_{t-1})}[p(y_t\,|\,x_t,\theta)]. \quad (2)$$

We model time-varying and path-dependent feasibility using a constraint state $z_t \in \mathcal{Z}$. This state can encode, for example, the previous design $x_{t-1}$, a physical system state, or the remaining budget. At time $t$, the admissible designs are given by a set $x_t \in \mathcal{X}(z_t) \subseteq \mathcal{X}$. The constraint state evolves according to a transition function $f : \mathcal{Z} \times \mathcal{X} \to \mathcal{Z}$ such that $z_{t+1} = f(z_t, x_t)$.

The goal is to find a deterministic policy $\pi : (\mathcal{D}_{t-1}, z_t) \mapsto x_t \in \mathcal{X}$ that proposes the optimal design $x_t = \pi(\mathcal{D}_{t-1}, z_t)$ given the observations and the constraints. Formally, the optimization problem is defined as:

$$\max_\pi \quad \mathbb{E}_{\theta \sim p(\theta),\, y_t \sim p(\cdot|x_t,\theta)} \left[ \textstyle\sum_{t=1}^T \text{EIG}(x_t; \mathcal{D}_{t-1}) \right] \quad (3)$$

$$\text{s.t.} \quad x_t = \pi(\mathcal{D}_{t-1}, z_t), \qquad t = 1,\dots,T,$$
$$x_t \in \mathcal{X}(z_t), \qquad t = 1,\dots,T,$$
$$z_{t+1} = f(z_t, x_t), \qquad t = 1,\dots,T,$$

where $y_{1:T} = (y_1,\dots,y_T)$ is then the observation sequence induced by the model under parameters $\theta$ by executing policy $\pi$ under constraint transitions $z_{t+1} = f(z_t, x_t)$ for $t = 1,\dots,T$. Note that in the unconstrained setting,

$\mathcal{X}(z_t) \equiv \mathcal{X}$ for all $t$, Equation (3) reduces to the standard finite-horizon objective optimized in (Foster et al., 2021). This formulation unifies several practically relevant constraint classes:

**Transition constraints** are local constraints that restrict the design at time $t$ as a function of the previous design (or, more generally, the current system/constraint state). A common example is a bounded-change constraint $\|x_t - x_{t-1}\| \le \delta$, which enforces that consecutive designs cannot differ by more than $\delta$ in a chosen norm. We express this by including the previous design in the constraint state (e.g., $z_t$ contains $x_{t-1}$), yielding the time-varying feasible set:

$$\mathcal{X}_t := \mathcal{X}(z_t) = \{x \in \mathcal{X} : \|x - z_t\| \le \delta\},$$

where $z_t = x_{t-1}$ and we assume an initial constraint state $z_1$ is given. Such constraints arise in practice due to actuator limits, smoothness requirements, or physical movement constraints (Verscheure et al., 2008; Paulson et al., 2022).

**Budget constraints** couple decisions across time by limiting the total resources that can be expended over the entire experimental campaign. We model this via a nonnegative *remaining budget* $b_t \in \mathbb{R}_{\ge 0}$, initialized at a total budget $B_{\text{total}}$, i.e., $b_1 = B_{\text{total}}$. The constraint state is then defined as $z_t = (\breve{z}_t, b_t)$, where $\breve{z}_t$ collects all *non-budget* constraint variables. Each experiment at design $x_t$ incurs a (possibly design- and state-dependent) cost $c_t := c(x_t, \breve{z}_t) \ge 0$, and the budget evolves deterministically as $b_{t+1} = b_t - c(x_t, \breve{z}_t)$. The admissible set is defined as:

$$\mathcal{X}_t = \{x \in \mathcal{X} : c(x, \breve{z}_t) \le b_t\},$$

Depending on the application, the number of experiments $T$ may be fixed or terminated when the budget is exhausted.

### 3.2. Multi-step Planning via Scenario Trees

The constrained objective in Equation (3) is challenging to optimize directly because constraints couple decisions across time via the evolving constraint state $z_t$ (e.g., remaining budget). We model the problem as a finite-horizon dynamic program (Bertsekas, 2012) over the state $(\mathcal{D}_{t-1}, z_t)$.

Let $V_t(\mathcal{D}_{t-1}, z_t)$ denote the optimal expected cumulative EIG from time $t$ onward. With reward $\text{EIG}(x; \mathcal{D}_{t-1})$, dynamics $y_t \sim p(\cdot \mid x_t, \theta)$, $\mathcal{D}_t = \mathcal{D}_{t-1} \cup \{(x_t, y_t)\}$, and $z_{t+1} = f(z_t, x_t)$, Bellman's principle of optimality yields the recursion (Bellman, 1956):

$$V_t(\mathcal{D}_{t-1}, z_t) = \max_{x_t \in \mathcal{X}(z_t)} \Big\{ \text{EIG}(x_t; \mathcal{D}_{t-1})$$
$$+ \gamma \mathbb{E}_{y_t \sim p(\cdot \mid x_t, \mathcal{D}_{t-1})} \big[ V_{t+1}(\mathcal{D}_t, z_{t+1}) \big] \Big\}, \quad (4)$$

with terminal condition $V_{T+1}(\mathcal{D}_T, z_{T+1}) := 0$. Here, $p(y_t \mid x_t, \mathcal{D}_{t-1})$ is the posterior predictive distribution defined in Equation (2) and $\gamma \in (0,1]$ is a discount factor

to down-weight the deeper-horizon returns. The resulting policy is $x_t^\star = \pi^\star(\mathcal{D}_{t-1}, z_t)$ given by the maximizer of Equation (4).

We approximate $V_t$ via receding-horizon planning: at time $t$, we solve an $H$-step truncated lookahead problem. The root of the tree corresponds to the current decision time $t$ and state $(\mathcal{D}_{t-1}, z_t)$. The tree alternates between choosing a design and branching over a finite set of fantasy observations $\tilde{y}$ sampled from the posterior predictive.

We index nodes by their *depth* $\ell \in \{0, \ldots, H\}$. A node at depth $\ell$ corresponds to decision stage $k = t + \ell$ and is identified by a branch index $j_{1:\ell} = (j_1, \ldots, j_\ell)$, which is the empty tuple $j_{1:0}$ at $\ell = 0$. Along any root-to-depth-$\ell$ path, for each simulated time $k = t, \ldots, t + \ell - 1$, we generate $m_k \in \mathbb{N}$ fantasy outcomes; the component $j_{k-t+1} \in \{1, \ldots, m_k\}$ records which branch is taken at time $k$. Thus, $j_{1:\ell}$ uniquely identifies a root-to-node path through the outcome branches. We apply the terminal value at depth-$H+1$, which we do not further expand.

For a *fixed* branch index $j_{1:\ell}$, we sample the fantasy outcomes $\{\tilde{y}_t^{j_1}, \tilde{y}_{t+1}^{(j_1, j_2)}, \ldots, \tilde{y}_{t+\ell-1}^{j_{1:\ell}}\}$. Let $(\mathcal{D}_{k-1}^{j_{1:\ell}}, z_k^{j_{1:\ell}})$ denote the dataset and constraint state obtained by sequentially updating from $(\mathcal{D}_{t-1}, z_t)$ with these outcomes. At the depth $\ell$ (time $k = t + \ell$), we select $x_k^{j_{1:\ell}} \in \mathcal{X}$; feasibility constraints will be enforced jointly across the tree in the optimization problem defined in Section 3.4. For $\ell < H$, we branch to the next depth by sampling $m_k$ fantasy outcomes from the posterior predictive,

$$\tilde{y}_k^{(j_{1:\ell}, j_{\ell+1})} \sim p(\cdot \mid x_k^{j_{1:\ell}}, \mathcal{D}_{k-1}^{j_{1:\ell}}), \quad j_{\ell+1} \in \{1, \ldots, m_k\}.$$

Each fantasy outcome defines a child node with an updated dataset and constraint state,

$$\mathcal{D}_k^{(j_{1:\ell}, j_{\ell+1})} = \mathcal{D}_{k-1}^{j_{1:\ell}} \cup \{(x_k^{j_{1:\ell}}, \tilde{y}_k^{(j_{1:\ell}, j_{\ell+1})})\},$$
$$z_{k+1}^{(j_{1:\ell}, j_{\ell+1})} = f\left(z_k^{j_{1:\ell}}, x_k^{j_{1:\ell}}\right).$$

Unrolling this construction up to decision depth $H$ yields a finite tree with decision variables $\{x_k^{j_{1:\ell}}\}$ at each decision node. Averaging over these fantasy branches leads to a Monte Carlo estimate of the expectation in Equation (4):

$$V_t^{(H)}(\mathcal{D}_{t-1}, z_t) = \max_{\mathbf{X}_{\text{tree}}}$$
$$\sum_{\ell=0}^{H} \left( \gamma^\ell \frac{1}{\prod_{r=0}^{\ell-1} m_{t+r}} \sum_{j_{1:\ell}} \text{EIG}\left(x_{t+\ell}^{j_{1:\ell}}; \mathcal{D}_{t+\ell-1}^{j_{1:\ell}}\right) \right), \quad (5)$$

where $\mathbf{X}_{\text{tree}} := \{x_{t+\ell}^{j_{1:\ell}}\}_{\ell=0}^{H}$ denotes all the design variables at decision nodes in the depth-$H$ tree and $\sum_{j_{1:\ell}}$ ranges over all branch indices at depth $\ell$ (with $j_{1:0}$ being the empty index at the root). By convention, $\prod_{r=0}^{-1} \cdot = 1$. At each time-step $t$, we use a receding-horizon strategy wherein we solve the

$H$-step scenario-tree problem of Equation (5) to get the optimized root design $x_t^\star$, observe $y_t$, update $(\mathcal{D}_t, z_{t+1})$, then repeat the lookahead planning at time $t + 1$.

### 3.3. Amortization for Scalable Tree Construction

While the scenario-tree formulation in Equation (5) removes the need to solve the nested recursion in Equation (4) explicitly, it still requires simulating fantasy outcomes and updating belief states at a large number of nodes, which is infeasible in the context of BED without additional simplifying assumptions or a conjugate posterior. We circumvent this computational issue by training a conditional density estimator, specifically, a mixture density network (Bishop, 1994), that learns an approximate mapping from the data to the posterior: $\mathcal{D} \mapsto p(\theta \,|\, \mathcal{D})$, thereby amortizing inference.

Let $q_\phi(\theta \,|\, \mathcal{D})$ denote a conditional density estimator with learnable parameters $\phi \in \Phi$. We estimate $\phi$ by minimizing the empirical negative log-likelihood loss:

$$\hat{\phi} := \arg\min_{\phi \in \Phi} -\frac{1}{n} \sum_{i=1}^{n} \log q_\phi(\theta_i \,|\, \mathcal{D}_i), \qquad (6)$$

using the training pairs $\{(\theta_i, \mathcal{D}_i)\}_{i=1}^{n}$ generated from the model. Specifically, we sample $\theta_i \sim p(\theta)$ and a dataset length $S_i \sim \mathrm{Unif}\{1, \ldots, T\}$, where $T$ is the maximum trajectory length. We then sample $\{x_{i,s}\}_{s=1}^{S_i} \sim p(x)$ from the design prior, simulate outcomes $y_{i,s} \sim p(\cdot \,|\, x_{i,s}, \theta_i)$. Once trained, we obtain an approximate posterior for dataset $\mathcal{D}$ by simply evaluating the estimator $q_{\hat{\phi}}(\theta \,|\, \mathcal{D}) \approx p(\theta \,|\, \mathcal{D})$. See Appendix A.2 for more details.

This inference network $q_\phi$ serves two critical roles. First, it enables rapid state updates: given a design $x$ and a fantasized outcome $\tilde{y}$, we update the belief $q_{\hat{\phi}}(\theta \,|\, \mathcal{D} \cup \{(x, \tilde{y})\})$ without retraining. Second, it facilitates sampling of fantasy outcomes from the posterior predictive distribution. At each node $k$ and for $j_{\ell+1} \in \{1, \ldots, m_k\}$, we sample:

$$\tilde{y}_k^{(j_{1:\ell}, j_{\ell+1})} \sim p(\cdot \,|\, x_k^{j_{1:\ell}}, \tilde{\theta}), \qquad \tilde{\theta} \sim q_{\hat{\phi}}(\cdot \,|\, \mathcal{D}_{k-1}^{j_{1:\ell}}). \quad (7)$$

**Efficient EIG estimation via** $q_\phi$**.** To evaluate the objective in Equation (5) and optimize candidate trees, we require a fast and differentiable surrogate for EIG at each node. At a node with history $\mathcal{D}$ (either realized $\mathcal{D}_t$ or a fantasized history along the scenario tree $\mathcal{D}_k^{j_{1:\ell}}$), we represent the current belief over $\theta$ by our amortized posterior $q_{\hat{\phi}}(\theta \,|\, \mathcal{D})$ and estimate EIG using an adaptive contrastive objective (Foster et al., 2020) (with $q_{\hat{\phi}}$ replacing the intractable posterior):

$$\widehat{\mathrm{EIG}}(x; \mathcal{D}, \hat{\phi}) := \mathbb{E}\left[ \log \frac{p(\tilde{y} \mid x, \theta_0)}{\frac{1}{L+1} \sum_{l=0}^{L} \frac{q_{\hat{\phi}}(\theta_l | \mathcal{D}) \, p(\tilde{y} | x, \theta_l)}{q_{\hat{\phi}}(\theta_l | \mathcal{D} \cup \{(x, \tilde{y})\})}} \right], \quad (8)$$

where the expectation is over: $\theta_0 \sim q_{\hat{\phi}}(\theta \,|\, \mathcal{D})$, $\tilde{y} \sim p(\cdot \,|\, x, \theta_0)$, and $\theta_{1:L} \sim q_{\hat{\phi}}(\theta \,|\, \mathcal{D} \cup \{(x, \tilde{y})\})$. When

$q_{\hat{\phi}}(\theta \,|\, \mathcal{D}) = p(\theta \,|\, \mathcal{D})$, Equation (8) becomes a valid lower bound on $\mathrm{EIG}(x; \mathcal{D})$ (Foster et al., 2020). Practically, it provides a *belief-state* EIG surrogate under $q_{\hat{\phi}}$, which is useful for fast test-time planning. When the likelihood is implicit, other estimators, e.g., Barber-Agakov (Foster et al., 2019) or critic-based estimators (Ivanova et al., 2021) can be substituted without changing the planning framework.

**Amortized policy for better initialization.** Optimizing all design variables $\{x_{t+\ell}^{j_{1:\ell}}\}$ over the scenario tree is a high-dimensional and non-convex problem, and naive random initialization can lead to poor local optima. We therefore warm-start the tree optimizer using a pre-trained, unconstrained, amortized design policy $\pi_\psi(x \,|\, \mathcal{D})$ as a proposal mechanism to generate high-quality initial trees. Such a strategy leverages the unconstrained policy $\pi_\psi$ to provide informative warm-starts that steer optimization toward high-information regions of the design space that are likely to remain competitive under the optimal constrained policy.

Concretely, for each decision node $(t + \ell, j_{1:\ell})$, we initialize $x_{t+\ell}^{j_{1:\ell}} \leftarrow \pi_\psi(\mathcal{D}_{t+\ell-1}^{j_{1:\ell}})$. In this work, we adopt the transformer-based policy of Huang et al. (2026), trained with reinforcement learning to maximize a variational lower bound (Foster et al., 2019) on the EIG. More details about ALINE are in Appendix A.3. Our method is agnostic to the specific choice of proposal policy, and alternative design policies (Shen et al., 2025; Bracher et al., 2026) can be used.

When constraints substantially shift the effective feasible region relative to the policy's training distribution, we adopt an exploration–exploitation initialization scheme (Powell, 2007; Huan & Marzouk, 2016): we initialize multiple trees either with $\pi_\psi$ (exploitation) or with a random policy (exploration). This hybrid initialization helps in cases where the constraints may cause the amortized policies to operate outside their training distribution, as we will see in Section 4.3.

### 3.4. One-shot Optimization via Reparameterization

Following the one-shot multi-step tree construction of Jiang et al. (2020b), we sample a collection of *base noise* variables $\varepsilon := (\varepsilon_\theta, \varepsilon_y)$ once and keep them fixed during optimization. Conditioning on $\varepsilon$ renders all fantasy outcomes deterministic functions of the tree decisions. Specifically, for each decision node $(k = t + \ell, j_{1:\ell})$, we draw posterior samples and fantasy observations via the reparameterization trick:

$$\theta_k^{j_{1:\ell}} = g_\phi\left( \mathcal{D}_{k-1}^{j_{1:\ell}}, \varepsilon_{\theta,k}^{j_{1:\ell}} \right),$$
$$\tilde{y}_k^{(j_{1:\ell}, j_{\ell+1})} = h\left( x_k^{j_{1:\ell}}, \theta_k^{j_{1:\ell}}, \varepsilon_{y,k}^{(j_{1:\ell}, j_{\ell+1})} \right).$$

Here $g_\phi$ is the sampling function induced by $q_{\hat{\phi}}(\theta \,|\, \mathcal{D})$, and $h$ generates observations from the likelihood model given $(x, \theta)$ and outcome noise.

Let $\widehat{V}^{(H)}(\mathbf{X}_{\mathrm{tree}}; \varepsilon)$ be the deterministic objective obtained

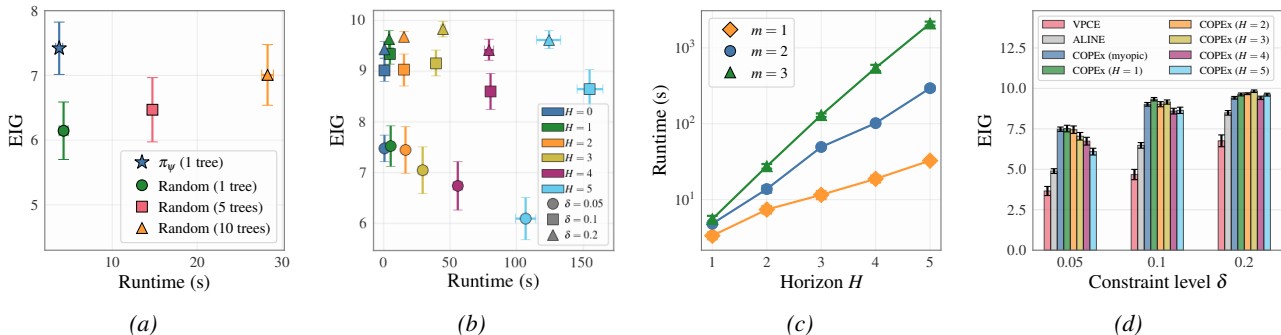

*Figure 3. Results on the location finding task.* **(a) Efficiency of amortized initialization.** We compare the design time and EIG of a single policy-initialized tree ($\pi_\psi$) against random multi-start initializations. **(b) Impact of planning horizon** $H$**.** We show EIG against design runtime across varying planning horizons $H$ and constraint levels $\delta$. Larger $H$ generally increases EIG at the cost of higher runtime, with diminishing returns after $H = 3$. **(c) Computational scaling.** Average optimization runtime per step (log scale) as a function of planning horizon $H$ and branching factor $m$. **(d) Performance comparison.** Final EIG against baselines across varying constraint levels $\delta$. COPEx consistently outperforms baselines. For each bar or point, we report the mean with 95% confidence intervals over 100 runs.

by evaluating Equation (5) from root state $(\mathcal{D}_{t-1}, z_t)$, replacing with the estimator in Equation (8), and generating all fantasy outcomes via the fixed base noise $\varepsilon$. Then, the constrained optimization problem becomes:

$$\mathbf{X}_{\text{tree}}^\star \in \arg\max_{\mathbf{X}_{\text{tree}}} \widehat{V}^{(H)}(\mathbf{X}_{\text{tree}}; \varepsilon) \qquad (9)$$

$$\text{s.t.} \quad x_k^{j_{1:\ell}} \in \mathcal{X}(z_k^{j_{1:\ell}}), \forall \text{ decision nodes,}$$

where $z_k^{j_{1:\ell}}$ are obtained by unrolling the dynamics $z_{k+1} = f(z_k, x_k)$ along each branch under $(\mathbf{X}_{\text{tree}}, \varepsilon)$. Transition constraints are enforced locally through $\mathcal{X}(z)$, while budget constraints are enforced by including the remaining budget in $z$ and updating it via $f$. We solve the resulting nonlinear program using sequential least squares programming (Kraft, 1988). After optimization, we execute the root design $x_t^\star = x_t^\emptyset$, observe $y_t$, update $(\mathcal{D}_t, z_{t+1})$, and re-plan at time $t + 1$.

## 4. Experiments

We now empirically validate COPEx on three complementary tasks under a variety of test-time constraints. In Section 4.1, we analyze and evaluate our method on the standard location finding BED benchmark subject to transition constraints on the design. In Section 4.2, we consider a global budget constraint within the constant elasticity of substitution (CES) task, a canonical problem in behavioral economics (Arrow et al., 1961). Finally, in Section 4.3, we show how COPEx can handle constrained active learning tasks under complex cost landscapes. Code to reproduce our experiments is available at https://github.com/yujiag21/COPEx.

### 4.1. Location Finding

The location finding task (Sheng & Hu, 2005) involves estimating the source of a signal in a two-dimensional design

space. In this setting, we analyze the effect of the bounded-change constraint that limits the maximum distance between consecutive designs to be $\delta = \{0.05, 0.1, 0.2\}$. The performance metric is the cumulative EIG estimate obtained after $T = 30$ steps. We begin by analyzing the effect of different choices and hyperparameters on our method. Detailed task specifications and experimental configurations are provided in Appendix B.1.

**Policy vs. random initialization of scenario tree.** We first examine the benefit of using the amortized policy $\pi_\psi$ to warm-start the tree optimizer, both in terms of cumulative EIG and the average time it takes to propose a design. Figure 3a compares a single $\pi_\psi$-initialized tree against multiple random policy initialized trees, with planning horizon $H = 1$. We observe that a single $\pi_\psi$-initialized tree achieves higher cumulative EIG than 10 randomly initialized trees, while taking significantly less time. Similar conclusions are drawn for $H = 2$, as shown in Appendix C.1. Even though $\pi_\psi$ is not trained to be constraint-aware, it still provides a high-quality feasible initialization in this case, which steers the constrained optimization solver towards better solutions.

**Impact of planning horizon** $H$**.** We next study how the planning horizon $H$ affects design quality vs. the computational time required to propose a design. In Figure 3b, we report the EIG against design runtime for $H \in \{0, \dots, 5\}$. Increasing $H$ generally improves EIG up to $H = 2$ or $H = 3$, beyond which additional planning yields little or no EIG gain despite substantially higher runtime. Given this trade-off, we adopt short horizons $H \in \{0, \dots, 3\}$ as a representative setting that balances EIG and computational cost for subsequent experiments.

**Computational cost.** As optimizing over all nodes of a large tree can increase computational cost, we analyze COPEx's performance versus its online runtime for differ-

ent planning horizons $H$ and number of branches $m_k$. For simplicity, we set the number of branches to be constant, i.e., $m_k = m$ for all $k$. In Figure 3c, we see that online runtime scales exponentially with the horizon depth. The impact of the number of branches on the runtime increases for larger horizons. In this task, we observe negligible performance improvements as $m$ increases; see Appendix C.1. We find that a small $m$ offers a favorable trade-off between performance and computational cost, and so we recommend setting $m = \{1, 2\}$, which we use for the remaining tasks.

**Performance against baselines.** Finally, we benchmark COPEx against two baselines: a non-amortized variational approach (VPCE) (Foster et al., 2020), and the state-of-the-art amortized policy ALINE (Huang et al., 2026). We evaluate COPEx with different horizons $H$, denoting the myopic variant ($H = 0$) as COPEx (myopic). We ensure VPCE satisfies the transition constraints via a differentiable reparameterization $x_t = x_{t-1} + \delta \cdot \tanh(x_u)$, where $x_u$ is the unconstrained optimization variable. This formulation guarantees $\|x_t - x_{t-1}\|_\infty < \delta$ while permitting standard gradient-based optimization. For ALINE, we restrict it to a post-hoc pool of feasible designs satisfying the constraint. We exclude comparison with continuous policy-based methods (Foster et al., 2021) which require non-trivial architectural changes or complex projection layers in order to satisfy design-dependent constraints.

Figure 3d shows that COPEx consistently outperforms both baselines across all constraint levels $\delta$, demonstrating the benefit of explicit constraint-aware planning. The performance difference between the baselines and COPEx increases as $\delta$ decreases, meaning that COPEx becomes more beneficial as the designs get more constrained.

## 4.2. Constant Elasticity of Substitution with Budget

Next, we evaluate our method on the Constant Elasticity of Substitution (CES) task. This task considers a behavioral economics problem in which a participant compares two baskets of goods and rates the subjective difference in utility between the baskets on a scale from 0 to 100. The design problem is to select pairs of baskets to infer the participant's latent utility parameters. In realistic elicitation settings, drastically altering the composition of goods between consecutive queries can induce cognitive overload or confuse the participants, e.g., by requiring a subject to mentally recalibrate from comparing small baskets of 5 items to bulk bundles of 200 items in a split second.

We consider a scenario where the total variation between consecutive designs in the whole trajectory is limited by a budget. The cost for each step $t$, representing the total change in the number of goods across all dimensions, is defined as $c_t = \sum_{i=1}^{d} |x_t^i - x_{t-1}^i|$, where $d$ is the dimension of the design space, and $x_t^i$ denotes the value of the $i$-th

*Table 1. Results on the CES task under global budget constraints. We report the EIG accrued before budget exhaustion for total budgets $B_{\text{total}} \in \{100, 150\}$. Values indicate mean $\pm$ 95% CI over 100 runs. COPEx with lookahead planning ($H \in \{1, 3\}$) consistently outperforms the myopic COPEx and all baselines.*

| Methods | $B_{\text{total}} = 100$ | $B_{\text{total}} = 150$ | Runtime (s) |
|---|---|---|---|
| COPEx (myopic) | 5.26±0.39 | 6.12±0.45 | 10.05±1.39 |
| COPEx ($H = 1$) | **7.03**±0.55 | **7.47**±0.55 | 19.47±1.90 |
| COPEx ($H = 3$) | 6.36±0.59 | 6.94±0.54 | 28.81±2.03 |
| VPCE | 2.18±0.02 | 2.54±0.25 | 105.75±3.03 |
| RL-BOED | 4.93±0.26 | 4.98±0.27 | 0.002±0.00 |
| ALINE | 4.46±0.21 | 5.70±0.24 | 0.07±0.01 |

dimension of the design vector at step $t$. We define the cumulative cost as $B_t = \sum_{s=1}^{t} c_s$. The experiment proceeds under the constraint $B_t \leq B_{\text{total}}$, terminating at step $T$ when the budget is exhausted. We conduct evaluations with total budgets of $B_{\text{total}} \in \{100, 150\}$, benchmarking COPEx against VPCE (Foster et al., 2020), RL-BOED (Blau et al., 2022), and ALINE (Huang et al., 2026). For all methods, we calculate the total EIG accrued before the budget exhausts.

The results in Table 1 show that COPEx with a lookahead horizon of $H = 1$ achieves the highest EIG, substantially outperforming its myopic counterpart ($H = 0$) and all the baselines across both budget levels, with only a modest runtime overhead relative to the myopic variant. While fully amortized baselines run in near-zero time, they do not explicitly optimize under a global budget constraint at test time and thus perform sub-optimally. Interestingly, increasing the horizon to $H = 3$ reduces performance. A plausible explanation is that deeper trees repeatedly query the amortized posterior, so any bias in $q_{\hat{\phi}}$ can compound along the rollout and eventually degrade decision quality.

## 4.3. Cost-aware Active Learning

**Transition constraint.** Our next experiment involves a cost-sensitive active learning task, where each design $x_t$ incurs a cost $c(x_t)$ determined by its location in the design space. We include two types of design-dependent cost, namely, Hazard Center and Rough Terrain; see Appendix B.3 for details. Additionally, we enforce a transition constraint on the maximum step size between consecutive designs, $\|x_t - x_{t-1}\|_\infty \leq \delta$, with $\delta = 1.0$. We conduct experiments on three standard benchmark functions: Ackley, Branin, and Goldstein-Price.

In this task, there are no explicit latent parameters $\theta$. The objective is to maximize the predictive accuracy at a set of held-out target inputs $x'_{1:M} \in \mathcal{X}_{\text{target}}$ with corresponding target values $y'_{1:M}$, while minimizing the cumulative cost of exploration. Therefore, instead of a parameter inference network, we train an amortized model to directly approximate

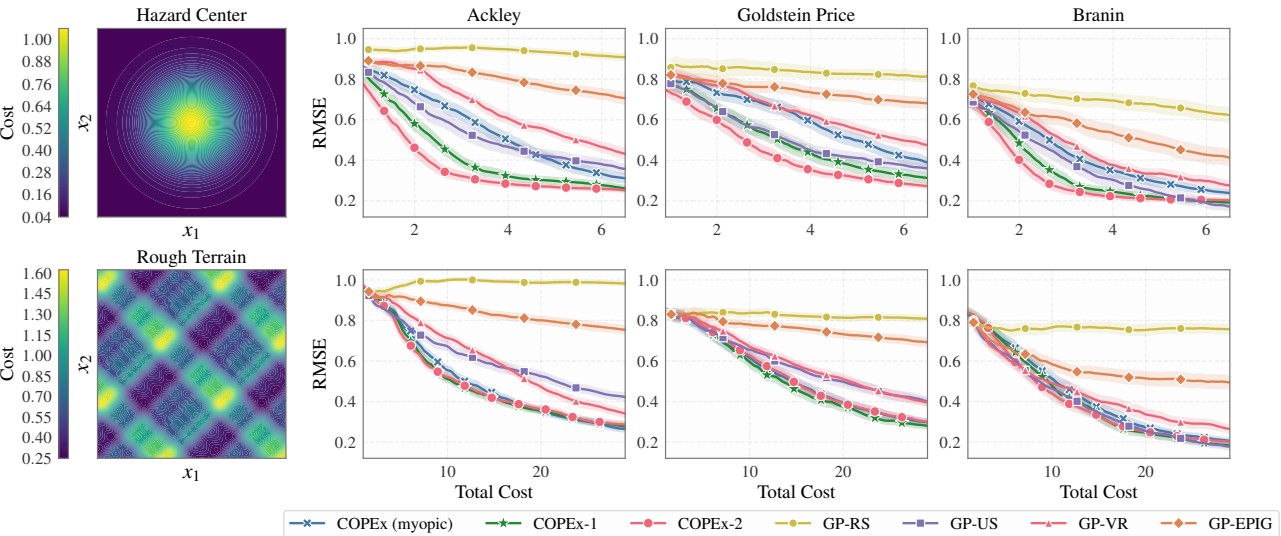

*Figure 4. Results on cost-aware active learning benchmark functions.* We compare the RMSE on a held-out test set against the cumulative cost incurred during exploration. **Rows:** Two cost landscapes (Top: Hazard Center; Bottom: Rough Terrain). **Columns:** Three benchmark functions (Ackley, Goldstein-Price, Branin). COPEx consistently achieves lower error than GP baselines for the same accumulated cost.

the posterior predictive distribution $q_\phi(y|x, \mathcal{D})$. We use the *Expected Predictive Information Gain* (EPIG) (Smith et al., 2023) as our utility, which measures the expected reduction in uncertainty regarding the target values $y'_{1:M}$:

$$\begin{aligned} \text{EPIG}(x; \mathcal{D}) :=\ & \mathcal{H}[p(y'_{1:M}|x'_{1:M}, \mathcal{D})] \\ & - \mathbb{E}_{y \sim p(y|x,\mathcal{D})} \left[ \mathcal{H}[p(y'_{1:M}|x'_{1:M}, \mathcal{D} \cup \{(x,y)\})] \right]. \end{aligned}$$

To incorporate heterogeneous costs, we maximize the cost-normalized EPIG, which is defined as:

$$\alpha(x_{t:t+H}; \mathcal{D}_{t-1}) = \frac{\sum_{\ell=0}^{H} \gamma^\ell \text{EPIG}(x_{t+\ell}; \mathcal{D}_{t+\ell-1})}{1 + w \sum_{\ell=0}^{H} \gamma^\ell c(x_{t+\ell})}, \quad (10)$$

where $w$ is the weight factor that controls the trade-off between maximizing EPIG and limiting the total cost. We estimate EPIG via Monte Carlo fantasies sampled from the amortized predictive surrogate. Full estimator and implementation details are given in Appendix B.3.

We compare both myopic and non-myopic COPEx against four non-amortized Gaussian Process (GP) baselines that select points by maximizing specific acquisition functions: Uncertainty Sampling (GP-US), Variance Reduction (GP-VR) (Yu et al., 2006), EPIG (GP-EPIG) (Smith et al., 2023), and Random Sampling (GP-RS). For a fair comparison, we adapt all acquisition functions to be cost-aware with the same factor of $1 + c(x)$, and restrict their candidate set to satisfy the transition constraint $\|x_t - x_{t-1}\| \le \delta$. We do not include any amortized AL method (e.g., ALINE (Huang et al., 2026) or AAL (Li et al., 2026)) in this task, as they cannot adapt to heterogeneous costs at test time without expensive retraining across different cost functions.

Figure 4 shows the root mean squared error (RMSE) on $M = 200$ target points as a function of the accumulated cost. COPEx consistently outperforms all GP baselines, achieving lower RMSE for the same total cost and faster convergence. The non-myopic COPEx generally outperforms the myopic variant, indicating the advantage of the lookahead planning in this task. This advantage is most pronounced under the Hazard Center cost, where COPEx-1 and COPEx-2 clearly separate from the baselines.

**Transition and budget constraints.** Finally, we study how COPEx performs in the cost-aware AL task under a fixed total budget $B$. The cost function and per-step transition constraint are the same as in the previous experiment, but the AL loop terminates once the accumulated cost $\sum_t c(x_t)$ reaches the budget $B$. We compare COPEx (myopic, $H{=}1$, and $H = 3$) against the same four continuously-optimized GP baselines. For this experiment, we adopt the model's predictive uncertainty, i.e., the predictive standard deviation of the target output, as the acquisition objective.

We consider two types of budgets: a small budget with respect to the planning horizon, where we expect planning to be more useful, and a large budget with respect to the planning horizon, where planning would only affect the last few design steps when the budget is about to be exhausted. Results in Figure 5 confirm this hypothesis: while COPEx outperforms all GP baselines in both settings, the benefit of planning depends on the remaining budget relative to horizon depth. In the small-budget case (Figure 5a, budgets of 2 and 4), longer planning horizons yield clear improvements over COPEx-myopic across all benchmarks: COPEx-

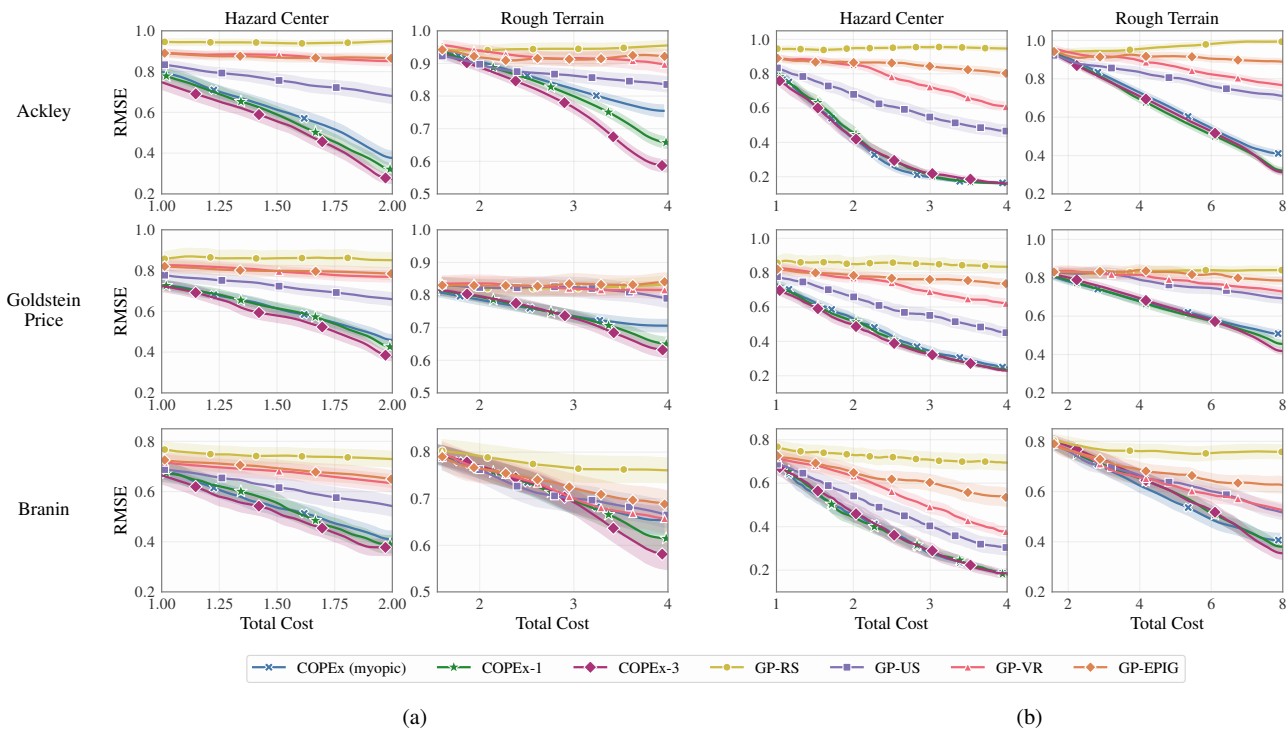

*Figure 5. Performance of COPEx and the GP baselines on the cost-aware active learning benchmarks under both transition and budget constraints.* **(a) Small-budget setting.** Results under budget of 2 for Hazard Center and 4 for Rough Terrain, corresponding to roughly 10 design steps before the budget exhausts. **(b) Large-budget setting.** Results under budget of 4 for Hazard Center and 8 for Rough Terrain, corresponding to roughly 20 design steps before the budget exhausts. Mean over 100 runs with 95% confidence intervals are shown.

3 achieves the lowest RMSE, followed by COPEx-1, with COPEx-myopic trailing both. However, in the large-budget setting with longer sequences (Figure 5b, budgets of 4 and 8), COPEx-myopic performs on par with the non-myopic variants, since the budget constrains only the final few steps and most of the trajectory is effectively unconstrained. This issue can be addressed as in Astudillo et al. (2021) via the use of a "fantasy" budget. At each step, they compute the remaining budget, choose a cheap base policy to simulate $N$ future evaluations, add up the $N$ fantasy costs, and use that as the current budget. Thus, they replace the full remaining budget with the estimated cost of $N$ base-policy future steps, which then influences the planning at each step.

## 5. Conclusion

We introduced COPEx, a novel method for Bayesian experimental design problems involving budget constraints and design-dependent feasibility limitations. By combining offline pre-training of posterior surrogates with online multi-step scenario tree search, COPEx enables non-myopic optimization under complex, test-time constraints. Moreover, COPEx is readily compatible with different acquisition criteria, such as EIG, EPIG, and predictive uncertainty.

**Limitations & future work.** COPEx depends on amortized

posterior networks for fast belief updates and fantasized rollouts. As a result, planning quality inherits approximation error from these surrogates, and errors can be amplified by long horizons as they accumulate along the rollout. More expressive conditional generative models (e.g., diffusion or flow-matching) are a promising avenue to reduce bias (Wildberger et al., 2023; Bracher et al., 2026). Second, explicit lookahead entails an inherent depth-breadth trade-off. In our experiments, COPEx incurs only modest, seconds-scale overhead for moderate horizons and branching. However, the underlying scenario tree still grows exponentially with lookahead depth. This motivates incorporating more advanced planning techniques to further improve efficiency (Somani et al., 2013; Cai et al., 2021). Third, our optimization is gradient-based and thus best suited to continuous design spaces; extending COPEx to discrete action spaces will require alternative optimizers (Bonami et al., 2011; Le Digabel, 2011). Finally, like most model-based approaches, our method relies on the assumption that the underlying model captures the true data-generating process. Addressing model misspecification remains a critical frontier; an exciting direction is to integrate our framework with recent robust methods (Huang et al., 2023; Forster et al., 2025; Tang et al., 2026).

## Acknowledgements

Y. Guo was supported by the Research Council of Finland grant no. 345604. D. Huang, S. Katt and S. Kaski were supported by the Research Council of Finland (Flagship programme: Finnish Center for Artificial Intelligence FCAI, 359207). S. Kaski was also supported by the UKRI Turing AI World-Leading Researcher Fellowship, [EP/W002973/1]. X. Zhang was supported by the Research Council of Finland (RCF-NSF FinBioFAB, grant agreement 365982). A. Bharti was supported by the Research Council of Finland grant no. 362534. This work is supported by ERC grant (ODD-ML 101201120). The authors wish to thank Aalto Science-IT project, and CSC–IT Center for Science, Finland, for the computational and data storage resources provided.

## Impact Statement

This paper presents work whose goal is to advance the field of Machine Learning. There are many potential societal consequences of our work, none of which we feel must be specifically highlighted here.

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

# Appendix

The appendix is organized as follows:

- In Appendix A, we provide comprehensive implementation details of COPEx, including the full algorithm, the architectures and training procedures for the posterior network and amortized policy, the hyperparameters for the optimizer, the vectorized scenario tree evaluation scheme and the analysis for planning error decomposition.

- In Appendix B, we elaborate on the experimental setup for each task, providing task descriptions, configurations for all baselines, and definitions of the evaluation metrics.

- In Appendix C, we present additional visualizations and ablation studies.

- In Appendix D, we provide an overview of the computational resources and software dependencies for this work.

## A. Additional Methodological and Implementation Details

### A.1. Full Algorithm

---

**Algorithm A1** COPEx planning procedure

---

1: **Input:** Task model $p(y \mid x, \theta)$, total experiments $T$, domain $\mathcal{X}$, planning horizon $H$, posterior network $q_{\hat{\phi}}$, policy $\pi_\psi$.
2: **Output:** Collected dataset $\mathcal{D}_T$.
3: **Initialize:** $\mathcal{D}_0 = \emptyset$, constraint state $z_1$.
4: **for** time $t = 1$ to $T$ **do**
5:      Initialize $V^{(H)} \leftarrow 0$
6:      **for** depth $\ell = 0$ to $H$ **do**
7:          Set $k = t + \ell$
8:          *// Start initialization of the tree and optimization objective.*
9:          **if** $\ell = 0$ **then**
10:             Initialize root node $x_k \leftarrow \pi_\psi(\mathcal{D}_{k-1})$ or uniform from $\mathcal{X}$
11:          **else**
12:             Initialize $m_{k-1}$ nodes, $\{x_k^{j_{1:\ell}}\}_{1:m_{k-1}} \leftarrow \pi_\psi(\mathcal{D}_{k-1}^{j_{1:\ell}})$ or uniform from $\mathcal{X}$
13:          **end if**
14:          **for** node $x_k^{j_{1:\ell}}$ at depth $\ell$ **do**
15:             **for** branch index $j_{\ell+1} = 1$ to $m_k$ **do**
16:                 **if** $\ell < H$ **then**
17:                     Fantasize $\tilde{y}_k^{(j_{1:\ell}, j_{\ell+1})}$ using Equation (7)
18:                     Set dataset $\mathcal{D}_k^{(j_{1:\ell}, j_{\ell+1})} \leftarrow \mathcal{D}_{k-1}^{j_{1:\ell}} \cup \{(x_k^{j_{1:\ell}}, \tilde{y}_k^{(j_{1:\ell}, j_{\ell+1})})\}$
19:                     Calculate constraint state $z_{k+1}^{(j_{1:\ell}, j_{\ell+1})} = f\left(z_k^{j_{1:\ell}}, x_k^{j_{1:\ell}}\right)$
20:                 **end if**
21:             **end for**
22:          **end for**
23:          Estimate one step $\text{EIG}_\ell = \frac{1}{\prod_{r=0}^{\ell-1} m_{t+r}} \sum_{j_{1:\ell}} \text{EIG}(x_k^{j_{1:\ell}}; \mathcal{D}_{k-1}^{j_{1:\ell}})$ (Equation (8))
24:          Update $V^{(H)} \leftarrow V^{(H)} + \gamma^\ell \text{EIG}_\ell$ (Equation (4))
25:      **end for**
26:      *// End initialization of the tree and optimization objective.*
27:      Optimize $\mathbf{X}_{\text{tree}}^\star \in \arg\max_{\mathbf{X}_{\text{tree}}} \hat{V}^{(H)}(\mathbf{X}_{\text{tree}}, \varepsilon)$ (Equation (9))
28:      Extract optimal root node $x_t^\star \leftarrow x_t^\emptyset \in \mathbf{X}_{\text{tree}}^\star$
29:      Execute $x_t^\star$ and observe $y_t$
30:      Update history $\mathcal{D}_t \leftarrow \mathcal{D}_{t-1} \cup \{(x_t^\star, y_t)\}$
31:      Update constraint state $z_{t+1} \leftarrow f(z_t, x_t^\star)$
32: **end for**
33: **return** $\mathcal{D}_T$

---

## A.2. Posterior Network

In this work, we train an inference neural network $q_\phi$ to serve as our inference model. We adopt the Transformer Neural Process (TNP) (Nguyen & Grover, 2022) structure designed to process a sequence of observed data points and map them to a posterior distribution modeled as a Gaussian Mixture Model (GMM), similar to the Amortized Conditioning Engine proposed in Chang et al. (2025).

**Architecture.** The network architecture consists of input embedding layers, a transformer encoder for processing dependencies, and a specialized inference head for probabilistic output. The data processing begins with embedding the inputs into a latent representation. The context set, consisting of observed design-outcome pairs $\mathcal{D}_t = \{(x_i, y_i)\}_{i=1}^t$, is processed by shared embedders. Specifically, designs $x_i$ and outcomes $y_i$ are passed through separate Multi-Layer Perceptrons (MLPs), denoted as $f_x$ and $f_y$, respectively, each consisting of a linear layer, a ReLU activation, and a final linear layer. The final embedding for a context point is obtained by summing these representations: $e_i = f_x(x_i) + f_y(y_i)$. To facilitate parameter inference, the model also receives a set of target tokens indicating the model parameters. These embeddings are then processed by the transformer layers. We employ a configuration with 3 transformer layers, each equipped with 4 attention heads. Within each layer, the feedforward networks have a dimension of 128, while the model's internal embedding dimension is maintained at 32 across all layers. The transformer utilizes self-attention mechanisms to model the dependencies within the context set $\mathcal{D}_t$ and cross-attention mechanisms to allow the target parameter embeddings to attend to the processed context representations. The output embeddings of the transformer corresponding to the parameter targets are fed into the inference head, which parameterizes the approximate posterior distribution. To capture potentially complex and multi-modal posteriors, we utilize a GMM with 10 components. The head consists of 10 separate MLPs, where each MLP processes the target embedding to output the parameters for one Gaussian component: a mixture weight, a mean, and a standard deviation.

To enable fully differentiable sampling from this mixture distribution for optimizing the scenario tree and fantasizing $\tilde{y}$, we employ a reparameterization strategy combining the Gumbel-Softmax trick with standard Gaussian reparameterization. Specifically, we do not sample a discrete component index; instead, we compute soft component probabilities $p_k$ using Gumbel-Softmax on the log-weights. Simultaneously, we generate reparameterized samples $\tilde{z}_k$ from every Gaussian component $k$ using standard noise $\epsilon \sim \mathcal{N}(0, 1)$. The final sample is obtained as the probability-weighted sum of component samples: $\theta = \sum_{k=1}^K p_k(\mu_k + \sigma_k \cdot \epsilon)$.

**Training.** For each training epoch, we generate a synthetic training pair $(\theta_i, \mathcal{D}_i)$ from the model. We first sample a ground-truth parameter $\theta_i \sim p(\theta)$ and a random dataset length $S_i \sim \mathrm{Unif}\{1, \ldots, T_{\max}\}$. We then sample a sequence of designs $\{x_{i,s}\}_{s=1}^{S_i} \sim p(x)$, and simulate the corresponding outcomes $y_{i,s} \sim p(\cdot|x_{i,s}, \theta_i)$ to form the dataset $\mathcal{D}_i := \{(x_{i,s}, y_{i,s})\}_{s=1}^{S_i}$. Training on random design sequences exposes the inference network to a diverse range of possible histories, ensuring robust posterior estimation regardless of the specific path taken by the planning algorithm during deployment. We fit the conditional density estimator $q_\phi(\theta|\mathcal{D})$ by minimizing the negative log-likelihood of the true parameters given the context data, i.e.,

$$\mathcal{L}(\phi) = -\mathbb{E}_{p(\theta)\, p(\mathcal{D}|\theta)}\big[\log q_\phi(\theta|\mathcal{D})\big], \qquad \hat{\phi} := \arg\min_{\phi \in \Phi} \widehat{\mathcal{L}}(\phi),$$

where $\widehat{\mathcal{L}}(\phi)$ denotes the empirical estimate using training pairs $\{(\theta_i, \mathcal{D}_i)\}_{i=1}^n$.

For predictive tasks targeting outcomes $y'$ at inputs $x'$, we analogously minimize the NLL under a target input distribution $p(x')$:

$$\mathcal{L}_p^{y'}(\phi) = -\mathbb{E}_{p(\theta)\, p(\mathcal{D}|\theta)\, p(x')\, p(y'|x',\theta)}\left[\sum_{m=1}^M \log q_\phi(y'_m|x'_m, \mathcal{D})\right].$$

This posterior-predictive objective is equivalent to the standard training objective of conditional neural processes (CNPs): maximizing the conditional log-likelihood of target outputs given a context set (here, $\mathcal{D}$).

**Implementation details.** The model is implemented using PyTorch. Training is performed using the AdamW optimizer with a cosine annealing learning rate schedule. The specific hyperparameters used for the inference architecture and training are detailed in Table A1. The training times of the posterior estimator for the location finding, CES, and active learning tasks are approximately 3, 2 and 4 hours, respectively.

*Table A1.* Model architecture and training hyperparameters for inference model

| Category | Parameter | Value |
|---|---|---|
| Architecture | Transformer layers | 3 |
| | Attention heads | 4 |
| | Embedding dimension | 32 |
| | Feedforward dimension | 128 |
| | GMM components | 10 |
| Training | Optimizer | AdamW |
| | Learning rate | 0.001 |
| | Weight decay | 0.01 |
| | Batch size | 200 |
| | Scheduler | Cosine Annealing |
| | Epochs | 100000 |

### A.3. Amortized Policy

We use the state-of-the-art amortized BED framework ALINE (Huang et al., 2026) as our policy network. ALINE unifies Bayesian inference and experimental design within a single TNP architecture (Nguyen & Grover, 2022). While the inference head approximates the posterior, a dedicated acquisition head concurrently learns a design policy $\pi_\psi(\mathcal{D}_t)$ that selects the most informative data points from a candidate pool $\mathcal{Q}$ to maximize information gain regarding a specified target. This reliance on a discrete candidate pool $\mathcal{Q}$ allows us to straightforwardly adapt the pre-trained policy to constrained experimental settings. By simply restricting the available actions to a valid subset $\mathcal{Q}_{\text{valid}} \subseteq \mathcal{Q}$ that satisfies current feasibility conditions, we can force the model to propose only valid designs from $\mathcal{X}(z_t, \mathcal{D}_{t-1})$. However, this naive application introduces a significant distribution shift: the policy, having been trained on unconstrained trajectories where all designs are feasible, encounters state distributions and history statistics during constrained deployment that differ substantially from its training distribution. Consequently, ALINE's direct policy rollout often yields suboptimal performance in strictly constrained scenarios.

**Architecture.** ALINE processes three sets of inputs: the historical context $\mathcal{D}_t$, the set of candidate queries $\mathcal{Q}$, and the target specifier $\xi$ (i.e., the indicator of $\theta$ in BED cases, and $x'$ in AL problems). To ensure the acquisition strategy is goal-oriented, ALINE employs a specific cross-attention mechanism where candidate query embeddings attend directly to the target embeddings. This allows the model to evaluate the relevance of each candidate design with respect to the specific inference goal. The processed query representations are then fed into the Acquisition Head, which consists of a 2-layer MLP with a hidden dimension of 128. The output is passed through a Softmax function to produce a valid probability distribution over the discrete query pool, from which the next experimental design $x_{t+1}$ is sampled.

**Training.** The policy is trained via reinforcement learning to maximize a variational lower bound on the total EIG (sEIG) over a trajectory. To overcome the challenge of sparse rewards in sequential design, ALINE utilizes a dense, per-step reward signal $R_t$ derived from its own inference network. Specifically, $R_t$ is defined as the immediate improvement in the log-probability of the approximate posterior (or predictive distribution) upon observing a new data point. The policy parameters $\psi$ are optimized using a policy gradient algorithm to maximize the expected cumulative discounted reward. To ensure stability, the policy and inference networks are trained jointly, often following a warm-up phase where only the inference network is trained to ensure reliable reward signals.

**Implementation details.** We report the specific hyperparameters used for the policy architecture and RL training in Table A2. The training times of the policy models for the location finding, CES, and active learning tasks are approximately 20, 13 and 12 hours, respectively.

### A.4. Additional Implementation Details

**Hyperparameters.** We solve the constrained optimization problem at the root of the tree using the Sequential Least Squares Programming (SLSQP) algorithm (Kraft, 1988). We use the hyperparameters listed in Table A3.

Regarding the node utility, to balance computational speed with accuracy during planning, we use the following sample

*Table A2.* Model architecture and training hyperparameters for policy model.

| Category | Parameter | Value |
|---|---|---|
| Architecture | Transformer layers | 3 |
| | Attention heads | 4 |
| | Embedding dimension | 32 |
| | Feedforward dimension | 128 |
| | GMM components | 10 |
| Training | Optimizer | AdamW |
| | Learning rate | 0.001 |
| | Weight decay | 0.01 |
| | Batch size | 200 |
| | Scheduler | Cosine annealing |
| | Epochs | 100000 |
| | Warm-up epochs | 20000 |
| | Query pool size | 200 |

sizes for the Monte Carlo estimators:

- EIG Estimation (ACE): We use $L = 32$ contrastive samples. For the nested expectations, we use 20 samples for the prior ($\theta_0$) and 20 samples for the outcome ($\tilde{y}$).

- EPIG Estimation: For the Active Learning task, we approximate the EPIG using 20 outcome samples ($\tilde{y}^*$) from the posterior predictive distribution.

**Vectorized scenario-tree evaluation.** To evaluate and differentiate the depth-$H$ scenario tree efficiently, we batch all nodes at the same depth and vectorize the rollout across outcome branches. Let $m_{t+\ell}$ denote the outcome branching factor at decision stage $t + \ell$ (depth $\ell$ from the root). We index nodes at depth $\ell$ by the Cartesian product

$$\mathcal{I}_\ell := \prod_{r=0}^{\ell-1}\{1, \ldots, m_{t+r}\}, \qquad \mathcal{I}_0 := \{\emptyset\},$$

so that each $i_{1:\ell} \in \mathcal{I}_\ell$ uniquely identifies a root-to-depth-$\ell$ outcome branch. We maintain the batched datasets and constraint states $\{\mathcal{D}_{t+\ell-1}^{i_{1:\ell}}, z_{t+\ell}^{i_{1:\ell}}\}_{i_{1:\ell} \in \mathcal{I}_\ell}$, and the batched decision variables $\{x_{t+\ell}^{i_{1:\ell}}\}_{i_{1:\ell} \in \mathcal{I}_\ell}$.

Given all designs at depth $\ell$, we generate fantasy observations in parallel:

$$\tilde{y}_{t+\ell}^{(i_{1:\ell}, j)} \sim p\big(y \mid x_{t+\ell}^{i_{1:\ell}}, \mathcal{D}_{t+\ell-1}^{i_{1:\ell}}\big), \qquad j \in \{1, \ldots, m_{t+\ell}\}.$$

Each fantasy outcome defines a child branch $(i_{1:\ell}, j) \in \mathcal{I}_{\ell+1}$, for which we update

$$\mathcal{D}_{t+\ell}^{(i_{1:\ell}, j)} \leftarrow \mathcal{D}_{t+\ell-1}^{i_{1:\ell}} \cup \{(x_{t+\ell}^{i_{1:\ell}}, \tilde{y}_{t+\ell}^{(i_{1:\ell}, j)})\},$$
$$z_{t+\ell+1}^{(i_{1:\ell}, j)} \leftarrow f\big(z_{t+\ell}^{i_{1:\ell}}, x_{t+\ell}^{i_{1:\ell}}\big).$$

In implementation, we represent the collection of nodes at depth $\ell$ as a single batch of size $|\mathcal{I}_\ell|$ and carry out posterior sampling, fantasy generation, and state updates using batched tensor operations. This enables efficient evaluation of the cumulative objective values $\{V_{t+\ell}^{i_{1:\ell}}\}_{i_{1:\ell} \in \mathcal{I}_\ell}$ (and their gradients) without explicit Python loops over branches.

**Hybrid Initialization Strategy.** In the active learning task, the introduction of an external cost function causes the test-time objective to shift significantly from the training objective. To improve the exploration, we adopt a mixed initialization strategy (Jiang et al., 2020a) by augmenting the policy-generated tree with 4 randomly generated trees for the optimization procedure. We optimize all five candidate trees and select the root design of the tree achieving the highest final utility.

*Table A3.* Optimization hyperparameters used in COPEx.

| Category | Parameter | Value |
|----------|-----------|-------|
| | Tolerance (`ftol`) | $1 \times 10^{-6}$ |
| | Max iterations (`maxiter`) | 600 |
| Optimization | Finite-difference step (`eps`) | $1 \times 10^{-4}$ |
| | Bounds tolerance | 0.008 |
| | Boundary penalty weight | 0.001 |

*Table A4.* Hyperparameters used in VPCE (Foster et al., 2020).

| Parameters | Location Finding | CES |
|------------|------------------|-----|
| VI gradient steps | 1000 | 1000 |
| VI learning rate | $10^{-3}$ | $10^{-3}$ |
| Design gradient steps | 2500 | 2500 |
| Design learning rate | $10^{-3}$ | $10^{-3}$ |
| Contrastive samples | 500 | 10 |
| Expectation samples | 500 | 10 |

### A.5. Planning Error Decomposition

COPEx approximates the optimal constrained BED policy through four successive relaxations of the ideal dynamic program: finite-horizon truncation (Equation (5)), amortized posterior estimation (Equation (6)), EIG estimation (Equation (8)), and non-convex tree optimization (Equation (9)). Let $J_t^\star$ denote the optimal constrained BED value from state $(\mathcal{D}_{t-1}, z_t)$, and let $J_t^{\text{COPEx}}$ denote the value induced by our $H$-step receding-horizon planner. Under bounded per-step utility $0 \le r_t \le R_{\max}$, the sub-optimality gap can be informally decomposed as:

$$J_t^\star - J_t^{\text{COPEx}} \le \epsilon_{\text{trunc}}(H) + \epsilon_{\text{post}}(H) + \epsilon_{\text{EIG}} + \epsilon_{\text{opt}},$$

where $\epsilon_{\text{trunc}}(H)$ is the finite-horizon truncation error, $\epsilon_{\text{post}}(H)$ is the error induced by replacing the exact posterior with the amortized surrogate $q_{\hat{\phi}}$, $\epsilon_{\text{EIG}}$ is the error from estimating Equation (8), and $\epsilon_{\text{opt}}$ is the optimization error from approximately solving the non-convex optimization problem in Equation (9).

With an $H$-step receding-horizon approximation, the lost tail value is bounded by $\epsilon_{\text{trunc}}(H) \le \sum_{\ell=H+1}^{T-t} \gamma^\ell R_{\max}$, which is at most $(T - t - H)R_{\max}$ when $\gamma = 1$. COPEx replaces exact Bayesian belief updates with the amortized posterior network $q_{\hat{\phi}}$, which can better approximate the true posterior as the number of simulated training pairs increases, provided the network is sufficiently expressive and optimization succeeds. In planning, if the induced one-step posterior predictive mismatch is uniformly bounded by $\epsilon_\phi$, then the induced error in the $H$-step lookahead value accumulates across the rollout and scales at most with the effective horizon: $\epsilon_{\text{post}}(H) = O(\sum_{\ell=0}^{H-1} \gamma^\ell \epsilon_\phi)$, which reduces to $O(H\epsilon_\phi)$ when $\gamma = 1$. This suggests that deeper lookahead can become less reliable when the amortized posterior is imperfect.

For EIG estimation, Equation (8) reduces to the lower bound of Foster et al. (2020) when $q_{\hat{\phi}}(\theta \mid \mathcal{D}) = p(\theta \mid \mathcal{D})$. In this case, the bound tightens monotonically with the number of contrastive samples $L$. When $q_{\hat{\phi}}(\theta \mid \mathcal{D}) \neq p(\theta \mid \mathcal{D})$, the objective in Equation (8) should instead be viewed as a surrogate utility whose deviation from the true EIG lower bound contains both posterior approximation bias and Monte Carlo error $\epsilon_{\text{MC}}$. Under finite-variance assumptions, $\epsilon_{\text{MC}}$ decreases at the usual $1/\sqrt{L}$ rate, while scenario-tree averaging over $m$ fantasy branches contributes an additional $1/\sqrt{m}$ dependence. Finally, $\epsilon_{\text{opt}}$ captures sub-optimality from approximately solving the non-convex one-shot tree problem, and depends on the optimization landscape, initialization, and solver budget.

## B. Experimental Details

### B.1. Location Finding

**Task description.** Location Finding is a standard benchmark in sequential Bayesian experimental design (Foster et al., 2019; 2021; Ivanova et al., 2021; Huang et al., 2026), where the goal is to infer the unknown locations of $K$ hidden sources in $\mathbb{R}^d$, $\theta = \{\theta_k \in \mathbb{R}^d\}_{k=1}^K$, by sequentially selecting measurement locations $x \in \mathbb{R}^d$ and observing noisy signal intensities. Each source emits a signal whose strength decays with distance according to an inverse-square law. Given a measurement at

*Table A5.* Hyperparameters used in RL-BOED ([Blau et al., 2022](#)).

| Parameters | CES |
|---|---|
| Critics | 2 |
| Random subsets | 2 |
| Contrastive samples | $10^5$ |
| Training epochs | $2 \times 10^4$ |
| Discount factor $\gamma$ | 0.9 |
| Target update rate | $5 \cdot 10^{-3}$ |
| Policy learning rate | $3 \cdot 10^{-4}$ |
| Critic learning rate | $3 \cdot 10^{-4}$ |
| Buffer size | $10^6$ |

location $x$, the total (unnormalized) signal intensity is modeled as a superposition of contributions from all sources:

$$\mu(\theta, x) = b + \sum_{k=1}^{K} \frac{\alpha_k}{m + \|\theta_k - x\|^2}, \tag{A1}$$

where $\alpha_k$ are known source strength constants, and $b, m > 0$ control the background level and the maximum signal intensity.

The observation is the log-transformed total intensity corrupted by Gaussian noise:

$$\log y \mid \theta, x \sim \mathcal{N}\big(\log \mu(\theta, x), \sigma^2\big). \tag{A2}$$

In our experiments, we follow the common setting with $K = 1$, $d = 2$, $\alpha_k = 1$, $b = 0.1$, $m = 10^{-4}$, and $\sigma = 0.5$. The design space is $x \in [0, 1]^d$, and the prior over the source location is factorized with each coordinate drawn uniformly $\theta_{k,j} \sim \mathrm{Unif}[0, 1]$, $j = 1, \ldots, d$.

**Baselines.** We compare COPEx with two baselines in this task:

- **VPCE** ([Foster et al., 2020](#)) maximizes the Prior Contrastive Estimation (PCE) lower bound via gradient ascent while updating the posterior using variational inference. The hyperparameters employed for VPCE are provided in Table A4.

- **ALINE** ([Huang et al., 2026](#)) is a policy-based BED method trained on the unconstrained version of the task, see Appendix A.3 for more details. We use the same set of hyperparameters as in Table A2.

**Evaluation details.** We evaluate performance using the Sequential Prior Contrastive Estimation (sPCE) lower bound ([Foster et al., 2021](#)) on the cumulative EIG. This is a standard metric in BED literature. For evaluation, we use a high-fidelity estimate with $L = 10^7$ contrastive samples.

### B.2. Constant Elasticity of Substitution

**Task description.** The Constant Elasticity of Substitution (CES) task ([Arrow et al., 1961](#)) is a canonical setting in economics for modeling a consumer's utility over bundles of goods. In our formulation, the experimenter seeks to infer latent preference parameters by querying the participant with comparisons between two baskets and observing a (possibly noisy) subjective preference signal.

A basket $z \in [0, 100]^K$ specifies quantities of $K$ goods. We denote the latent preference parameters by $\theta = (\rho, \boldsymbol{\alpha}, u)$, where $\rho \in (0, 1)$ controls substitutability across goods, $\boldsymbol{\alpha} \in \Delta^{K-1}$ is a simplex-valued weight vector over goods, and $u > 0$ scales the overall sensitivity of the response. Each design (query) consists of a pair of baskets $x = (z, z') \in [0, 100]^{2K}$, and the observed outcome is a reported preference strength $y \in [0, 1]$.

The utility of a basket is defined by the CES utility function: $U(z) = \left(\sum_{i=1}^{K} z_i^\rho \alpha_i\right)^{\frac{1}{\rho}}$. We place the following prior over the latent parameters:

$$\rho \sim \mathrm{Beta}(1, 1), \qquad \boldsymbol{\alpha} \sim \mathrm{Dirichlet}(\mathbf{1}_K), \qquad \log u \sim \mathcal{N}(1, 3^2). \tag{A3}$$

Given a query $x = (z, z')$ and parameters $\theta$, we model the latent (noisy) utility difference as

$$\eta \sim \mathcal{N}\Big( u\left(U(z) - U(z')\right),\ u^2\,\tau^2\,(1 + \|z - z'\|)^2\Big), \tag{A4}$$

and map it to the bounded reported outcome using a logistic link followed by clipping $y = \text{clip}\big(\sigma(\eta),\ \epsilon,\ 1 - \epsilon\big)$, where $\sigma(\cdot)$ denotes the sigmoid function. Throughout our experiments, we set $K = 3$, $\tau = 0.005$, and $\epsilon = 2^{-22}$.

**Baselines.** We compare COPEx with three baselines in this task:

- **RL-BOED** (Blau et al., 2022) formulates the design policy optimization as a Markov Decision Process (MDP) and employs reinforcement learning to learn the optimal strategy. It utilizes a stepwise reward function to estimate the marginal contribution of each experiment to the total information gain. The policy is trained using Randomized Ensembled Double Q-learning. Detailed hyperparameters for RL-BOED are listed in Table A5.

- **VPCE** (Foster et al., 2020) is implemented as described in Appendix B.1, with hyperparameters listed in Table A4.

- **ALINE** (Huang et al., 2026) is implemented as described in Appendix B.1. We use the same hyperparameters as in Table A2.

**Evaluation details.** We report the cumulative EIG accrued until the budget is exhausted. As with the location finding task, we approximate the EIG using the sPCE estimator with $L = 10^7$ samples.

### B.3. Active Learning

**Task description.** We consider an active learning task on a 2D domain $\mathcal{X} = [-5, 5]^2$ where the goal is to maximize predictive accuracy on a held-out test set under spatially varying costs. We use two distinct cost landscapes:

- **Hazard Center:** A radial cost function peaking at the origin, simulating a central hazard or resource-intensive zone. It is modeled as a Gaussian bump: $c(\mathbf{x}) = c_{\min} + A \exp(-\|\mathbf{x}\|^2 / 2\sigma^2)$. We set the base cost $c_{\min} = 0.05$, peak amplitude $A = 1.0$, and width $\sigma = 1.5$. The weight factor $w$ is 1.

- **Rough Terrain:** A multi-frequency terrain simulating complex environments like uneven planetary surfaces. It is modeled as a superposition of sinusoidal waves passed through a Softplus activation to ensure non-negativity: $c(\mathbf{x}) = \text{Softplus}(\sum a_i \sin(u_i) \cos(v_i)) + \epsilon$. We sum three wave components ($i = 1, 2, 3$) with decreasing amplitudes $\mathbf{a} = \{1.0, 0.5, 0.25\}$, increasing frequencies $\mathbf{k} = \{1.0, 2.0, 4.0\}$, and phase shifts $\phi = \{0.3, 1.1, 2.2\}$ and $\psi = \{1.5, 0.7, 2.8\}$. The weight factor $w$ is 3 for this function.

In addition to the cost landscapes, we evaluate performance on three standard synthetic benchmark functions serving as the ground truth target $f(\mathbf{x})$. We linearly rescale the inputs from their original domains to our experimental design space.

- **Ackley Function.** A widely used multimodal test function characterized by a nearly flat outer region and a large hole at the center:

$$\text{Ackley}(\mathbf{x}) = -a \exp\left(-b\sqrt{\frac{1}{d_{\mathcal{X}}}\sum_{i=1}^{d_{\mathcal{X}}} x_i^2}\right) - \exp\left(\frac{1}{d_{\mathcal{X}}}\sum_{i=1}^{d_{\mathcal{X}}} \cos(c x_i)\right) + a + \exp(1), \tag{A5}$$

where $a = 20$, $b = 0.2$, and $c = 2\pi$.

- **Branin Function.** A function with three global minima, often used to test the ability of global optimization algorithms:

$$\text{Branin}(x_1, x_2) = a(x_2 - b x_1^2 + c x_1 - r)^2 + s(1 - t)\cos(x_1) + s, \tag{A6}$$

where $a = 1$, $b = \frac{5.1}{4\pi^2}$, $c = \frac{5}{\pi}$, $r = 6$, $s = 10$, and $t = \frac{1}{8\pi}$.

- **Goldstein-Price Function.** A complex function with many local minima, suitable for testing exploration capabilities. We use the logarithmic form:

$$
\text{Goldstein-Price}(x_1, x_2) = \frac{1}{2.427} \Bigg[ \log \Bigg( \big[1 + (x_1 + x_2 + 1)^2 (19 - 14x_1 + 3x_1^2 - 14x_2 + 6x_1 x_2 + 3x_2^2)\big]
$$
$$
\cdot \big[30 + (2x_1 - 3x_2)^2 (18 - 32x_1 + 12x_1^2 + 48x_2 - 36x_1 x_2 + 27x_2^2)\big] \Bigg) - 8.693 \Bigg] \tag{A7}
$$

**Active learning under budget constraints.** For this experiment, we use the model's predictive uncertainty as the acquisition objective, based on the uncertainty-sampling baseline (GP-US). Given the ALINE posterior predictive distribution represented as a Gaussian mixture,

$$
p(y \mid x, \mathcal{D}) = \sum_{k=1}^{K} w_k(x) \mathcal{N} \left( y \mid \mu_k(x), \sigma_k^2(x) \right),
$$

where $\{w_k, \mu_k, \sigma_k\}$ are the mixture weights, means, and standard deviations of the predictive distribution at $x$. We compute the predictive mean as

$$
\mu_{\text{pred}}(x) = \sum_{k=1}^{K} w_k(x) \mu_k(x),
$$

and the predictive variance using the law of total variance,

$$
\sigma_{\text{pred}}^2(x) = \sum_{k=1}^{K} w_k(x) \left( \sigma_k^2(x) + \mu_k^2(x) \right) - \left( \sum_{k=1}^{K} w_k(x) \mu_k(x) \right)^2.
$$

The acquisition value is then the predictive standard deviation

$$
a(x) = \sigma_{\text{pred}}(x).
$$

**Baselines.** We compare against four Gaussian Process (GP) baselines: random, uncertainty, variance-reduction, and EPIG acquisition. To make these baselines cost-aware, we maximize the cost-normalized objective $\frac{\alpha(x)}{1 + \lambda c(x)}$ with continuous optimization, where $\lambda$ is a cost-weighting factor (set to 1) and $\alpha(x)$ is the original acquisition function, including uncertainty, variance-reduction, and EPIG. The transition constraint is enforced directly through the optimizer bounds: at each step we maximize the acquisition function within the $\ell_\infty$ box of half-width $\delta$ centred on the previously selected design, intersected with the global domain $[-5, 5]^d$. We solve this constrained problem with SLSQP via `scipy.optimize.minimize`, using 5 random restarts and a maximum of 100 iterations per restart, retaining the best feasible solution. Each selected design is labelled by querying the true objective. The GP uses a constant-scaled RBF kernel with observation noise $\alpha = 10^{-8}$.

- **Random Sampling (GP-RS):** As a baseline for lower-bound performance, we employ a random policy that selects the next query point $x$ uniformly at random from the available candidate pool $\mathcal{Q}$.

- **Uncertainty Sampling (GP-US):** This is a widely adopted strategy that prioritizes acquiring data points where the model's predictive uncertainty is highest. It is defined simply as the predictive standard deviation at a candidate input $x$:

$$
\alpha_{\text{US}}(x) = \sqrt{\mathbb{V}[y \mid x, \mathcal{D}]},
$$

where $\mathbb{V}[y \mid x, \mathcal{D}]$ denotes the predictive variance given the current dataset $\mathcal{D}$.

- **Variance Reduction (GP-VR)** (Yu et al., 2006): Variance Reduction seeks to select the candidate point that maximizes the expected reduction in aggregate predictive variance across a specific set of test points $\{x'_m\}_{m=1}^{M}$. The acquisition function is defined as:

$$
\alpha_{\text{VR}}(x) = \sum_{m=1}^{M} \frac{\text{Cov}_{\text{post}}(x'_m, x)^2}{\mathbb{V}[y \mid x, \mathcal{D}]},
$$

where $\mathrm{Cov}_{\mathrm{post}}(x', x)$ represents the posterior covariance between the latent function values at the test point $x'$ and the candidate $x$, conditioned on the history $\mathcal{D} = \{(X_{\mathrm{train}}, y_{\mathrm{train}})\}$. This covariance is computed using the kernel function $k(\cdot, \cdot)$ and the noise variance $\alpha$ as:

$$\mathrm{Cov}_{\mathrm{post}}(x', x) = k(x', x) - k(x', X_{\mathrm{train}})(K_{\mathrm{train}} + \alpha I)^{-1} k(X_{\mathrm{train}}, x), \qquad K_{\mathrm{train}} = k(X_{\mathrm{train}}, X_{\mathrm{train}}).$$

- **Expected Predictive Information Gain (GP-EPIG)** (Smith et al., 2023): This objective measures the expected reduction in uncertainty regarding the predictive distribution on a target input distribution $p_*(x^*)$. Assuming a Gaussian predictive distribution, EPIG can be formulated as the expectation over the target distribution:

$$\alpha_{\mathrm{EPIG}}(x) = \mathbb{E}_{p(x')} \left[ \frac{1}{2} \log \left( \frac{\mathbb{V}[y \mid x, \mathcal{D}] \mathbb{V}[y' \mid x', \mathcal{D}]}{\mathbb{V}[y \mid x, \mathcal{D}] \mathbb{V}[y' \mid x', \mathcal{D}] - \mathrm{Cov}_{\mathrm{post}}(x', x)^2} \right) \right]$$

**Evaluation details.** We evaluate performance using the Root Mean Squared Error (RMSE) of the predictive mean on a held-out set of 200 randomly sampled target points. We report the RMSE as a function of the total accumulated cost.

## C. Additional Experimental Results

### C.1. Location Finding

**Policy vs. random initialization of scenario tree.** We observe that a single $\pi_\psi$-initialized tree achieves higher EIG than randomly initialized trees with a planning horizon of $H = 1$. This trend persists for deeper horizons $H = 2$, although the performance gap between policy and random initialization narrows (see Figure A1a). We observe that for $H = 2$, increasing the number of random restart trees degrades final performance. This may be caused by the error of the inference model or the Monte Carlo estimator. In the higher-dimensional optimization landscape, generating more random candidates increases the likelihood of selecting a trajectory that overfits to the error, rather than one that is genuinely informative.

**Branching factor analysis.** We analyze the choice of the branching factor vs. EIG in Figure A1b with $\delta = 0.05$. We use the same $m$ for the whole $H$ horizon. The choice of $m$ presents a trade-off between estimation variance and computational cost. When the branch number increases, such as $m = 3$, the planning time increases, but does not outperform $m = 2$ in terms of Final EIG. Increasing $m$ theoretically improves the quality of the lookahead estimator by aggregating more fantasized observations $y$ to reduce variance. However, this comes at a cost: it exponentially increases the dimensionality of the decision space (the number of $x$ variables) in the joint optimization problem. This expansion can destabilize the optimization, as the solver must navigate a significantly more complex, non-convex landscape. Consequently, we select $m = 1$ and 2 as they offer the best balance of computational feasibility and robust performance. We choose $m = 2$ for the location finding task, $m = 1$ for the CES and active learning tasks.

**Varying discount factor $\gamma$.** We set $\gamma \in \{0.5, 0.8, 0.9, 1.0\}$ and measure the EIG in each case under the planning horizon of $H = \{1, 2\}$. Results in Figure A1c indicate that the choice of $\gamma$ has little impact on the cumulative EIG for both planning horizons. Given this empirical robustness, we fix $\gamma = 0.8$ as a representative value for all subsequent experiments to maintain consistency.

**Trajectory under different constraints and planning horizon.** We show the design trajectory of COPEx under constraints $\delta = 0.05$ (Figure A2) and 0.1 (Figure A3) on the same ground truth. This visualization illustrates the different behavior of COPEx under different constraints, which iteratively selects query points towards the target area.

**Posterior approximation error accumulation along fantasy rollouts.** Since both fantasy outcome generation (Equation (7)) and EIG estimation (Equation (8)) in our planning framework rely on the amortized posterior $q_{\hat{\phi}}$, its accuracy drives the quality of the scenario tree. We empirically quantify how the posterior approximation error accumulates with the planning horizon. On the location finding task, we use importance sampling (IS) to obtain a high-fidelity reference posterior. We run two fantasy rollout pipelines using identical designs at each depth: one using our amortized posterior $q_{\hat{\phi}}$, and one using the IS-based posterior. At each depth $\ell$, we measure the Maximum Mean Discrepancy (MMD; Gretton et al. (2006)) between the posterior samples produced by the two pipelines, averaged over 100 independent rollouts. As shown in Table A6, the posterior divergence increases monotonically with depth, confirming that approximation errors from the amortized surrogate compound along fantasy rollouts. This is consistent with the error decomposition discussed in the paper and indicates that deeper lookahead becomes more sensitive to surrogate error.

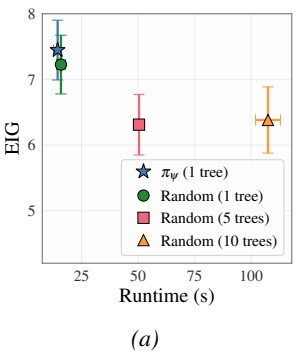 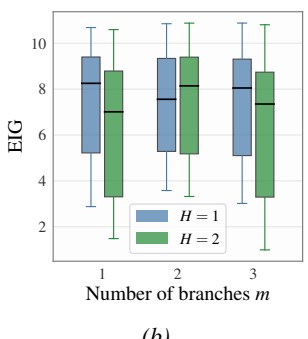 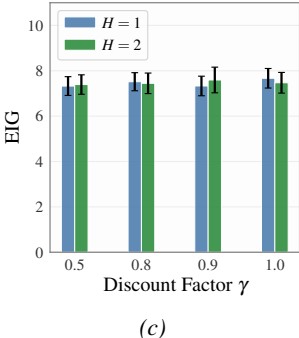

| (a) | (b) | (c) |

*Figure A1. Additional results for location finding task.* Design time vs. EIG of a single policy-initialized tree against random multi-start initializations (with $\delta = 0.05$, $H = 2$). (b) EIG vs. branching factor $m \in \{1, 2, 3\}$ at each tree level (with $\delta = 0.05$). (c) Impact of discount factor $\gamma$. Cumulative EIG across varying $\gamma$. Performance is largely insensitive to $\gamma$.

*Table A6.* Maximum Mean Discrepancy (MMD, $\pm$SE) between posterior samples from the amortized-posterior rollout and the importance-sampling reference rollout, as a function of fantasy depth $\ell$, on the location finding task. Lower is better; values are averaged over 100 independent rollouts.

| Metric ($\pm$SE) | $\ell = 0$ | $\ell = 1$ | $\ell = 2$ | $\ell = 3$ | $\ell = 4$ | $\ell = 5$ |
|---|---|---|---|---|---|---|
| MMD | 0.42$\pm$0.004 | 0.48$\pm$0.004 | 0.53$\pm$0.004 | 0.58$\pm$0.003 | 0.60$\pm$0.004 | 0.64$\pm$0.004 |

## C.2. Active Learning

**Trajectory analysis.** Figure A4 illustrates the trajectory in Section 4.3 under the Hazard Center cost function. We observe a distinct cost-aware strategy: the agent initially prioritizes exploring the low-cost peripheral regions. Only after reducing uncertainty at the boundaries does it transition to the high-cost region, balancing information gain against the varying cost.

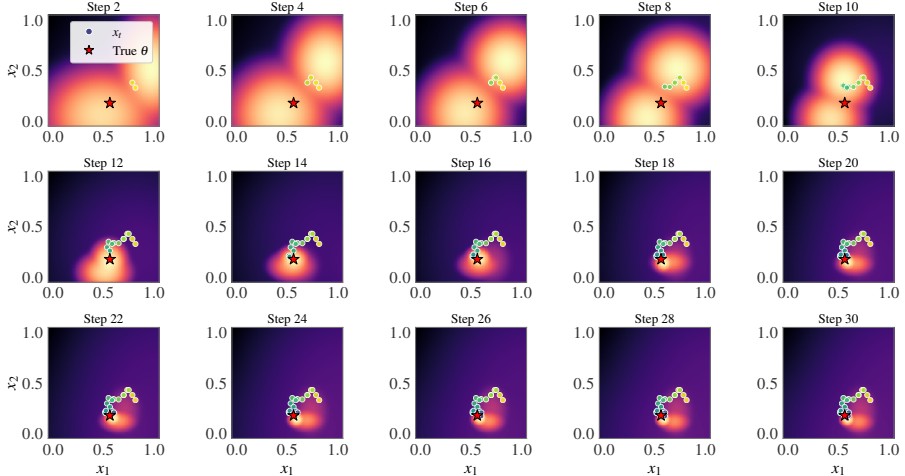

*Figure A2.* An example of the sequential query strategy of COPEx on location finding task over 30 steps under $H = 1, \delta = 0.05$.

## D. Computational Resources and Software

All experiments conducted in this work, encompassing model development, hyperparameter optimization, baseline evaluations, and preliminary analyses, are performed on a GPU cluster equipped with NVIDIA Tesla V100-SXM2 32GB or A100-SXM4-80GB GPUs. The total computational resources consumed for this research, including all development stages and experimental runs, are estimated to be approximately 5000 GPU hours. The core code base is built using PyTorch (https://pytorch.org/, License: modified BSD license). For the Gaussian Process (GP) based baselines, we utilize Scikit-learn (Pedregosa et al., 2011) (https://scikit-learn.org/, License: modified BSD license). Our

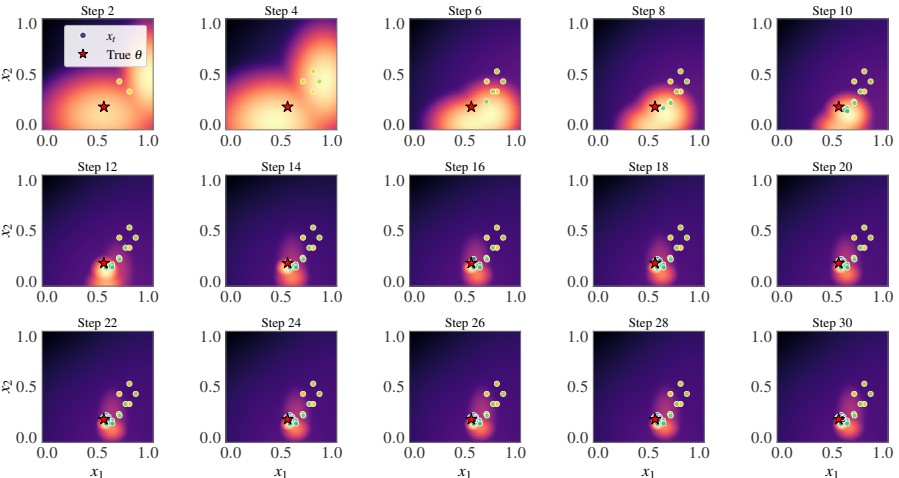

*Figure A3.* An example of the sequential query strategy of COPEx on location finding task over 30 steps under $H = 1, \delta = 0.1$.

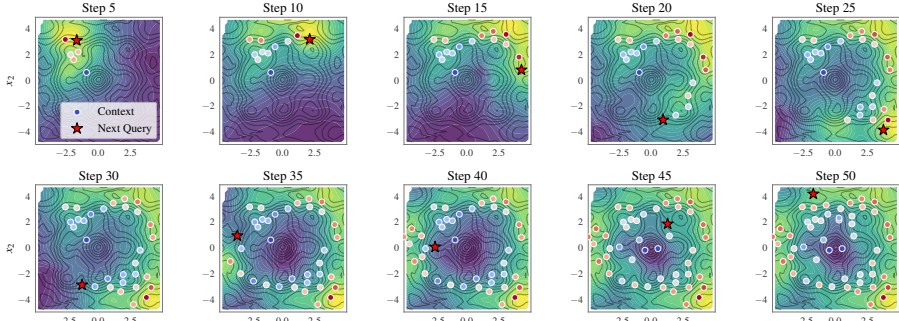

*Figure A4.* An example of the sequential query strategy of COPEx on Ackley function over 50 steps under $H = 1, \delta = 1$.

implementations of VPCE, RL-BOED, and ALINE are adapted from their official repositories.

