# OpenReview forum: "Constrained Bayesian Experimental Design via Online Planning"
_ICML.cc/2026/Conference — ICML 2026 regular_

### Official Review · Reviewer_1v3e · 2026-03-13

**Soundness:** 3
**Presentation:** 3
**Significance:** 3
**Originality:** 3
**Overall Recommendation:** 4
**Confidence:** 4

**Summary:**

This paper considers (sequential) Bayesian experimental design (BED), in which there is some uncertain state of the world $\theta$, and an ability to observe datapoints $y_t$ whose likelihood is a known function of some $x_t$ and $\theta$.  There is a prior probability distribution over $\theta$. The goal is to minimize the entropy of the posterior distribution on $\theta$ following a sequence of adaptively-chosen $x_t$.  The $x_t$ are subject to dynamic constraints, such as motion constraints ($x_t$ is the robot's location and the robot can only move so far in each timestep) or budget constraints on the total cost of experiments performed.

The paper proposes pre-training a novel inference network for producing the posterior distribution from a history of data. Within a multi-step tree-based rollout scheme, it uses this inference network to perform efficient estimation of the expected information gain (expected reduction in posterior entropy).  It also proposes a novel warm-starting technique based on a previously-published method for finding good policies for BED in unconstrained settings.  Following previous work on multi-step tree-based rollout for Bayesian optimization (Jiang et al. 2020), it also proposes optimizing the policy using one-shot optimization over the tree, leveraging the reparameterization trick adapted to their inference network.

Experiments are presented demonstrating the effectiveness of their method.

**Compliance With Llm Reviewing Policy:**

Affirmed.

**Final Justification:**

This paper presents a nice technical contribution that I feel will be helpful in a variety of problems. The most significant limitation in my mind is that computation scales exponentially with the horizon.  The rebuttal was useful in helping me to understand this limitation more fully and I hope my suggestions will be helpful to the author in improving their paper.

**Key Questions For Authors:**

- Perhaps I missed it, but I did not see the time required for pre-training. This is not included in the runtime results reported in the main paper --- am I understanding that correctly?

- My understanding is that the improvement offered by going from myopic to multi-step lookahead is not especially large. In some cases it helps noticeably, but in a number of problems the performance is almost the same. This would seem to reduce the significance of developing a multi-step method for this setting.  Could the authors comment on whether the myopic variant of their algorithm is also new?

- One of the proposed applications of the method is to budgeted settings where the budget limits what experiments can be performed.  I'd like to understand how the algorithm behaves in a setting where I want to maximize EIG subject to a budget on the total cost of all observations.  Given that the planning horizon H is typically short (1 to 3 steps ahead), is it correct that the algorithm would only take the budget into account when it can be completely consumed within H steps?  I'd like to make sure that I understand the implications of this.  Suppose I have a problem like the cost-aware active learning problem in 4.3 where the budget is B=100 and the cost of an observation is 1 or c=2.  In general, we'd expect a good algorithm to prefer choosing a lower-cost observation, assuming that its other properties are the same.  But in this setting, the proposed algorithm would only take the observation cost into account if the remaining budget is less than c*H.  Is that correct?  It seems like one could use the phenomenon to design problems where the proposed algorithm performs quite poorly relative to other baselines that can account for the observation cost more fully.

- In light of the previous comment, I'm wondering what would happen if you changed the objective in the cost-aware active learning problem from the normalized EPIG to just the EPIG, and constrained the total cost. Would it still perform well relative to baselines?  And is it correct that the one-step reward in the problem as implemented is the change in normalized EPIG?

**Limitations:**

The paper has a nice limitations discussion in the conclusion.  But if the answers to my other questions reveal additional limitations beyond what is discussed there, it should be added.

**Strengths And Weaknesses:**

Strengths
- I found the paper well-written.  The standard model in this area is difficult to convey both clearly and correctly, but the paper does a good job with this
- The approach taken is well-grounded and appropriate for the problem
- The method outperforms baselines on a reasonable set of problems
- IMO the level of novelty is is high enough for ICML.  It builds substantially on previous work, but combines the pieces in interesting and useful ways

Weaknesses
- As is typical with multi-step tree methods, computation scales exponentially with the horizon
- Perhaps for this reason, the algorithm uses only short horizons (H=1,2,3) in the experiments.  Several of the experiments also use a short horizon.

UPDATE AFTER REBUTTAL
Thanks to the authors for their rebuttal.  I found it helpful.  I continue to feel that this paper is a weak accept.

---

> ### Author Rebuttal · Authors · 2026-03-30
>
> Thank you for your positive feedback on our work and for the series of insightful questions. We address each of them in detail below.
>
> > *1. Perhaps I missed it, but I did not see the time required for pre-training. This is not included in the runtime results reported in the main paper --- am I understanding that correctly?*
>
> Apologies for this oversight. Here are the pre-training times on a GPU cluster equipped with NVIDIA Tesla V100, which we will include in the appendix:
>
> |  | Location finding | CES | Active learning  |
> | :---- | :---- | :---- | :---- |
> | Posterior estimator | 3h | 2h | 4h |
> | policy | 20h | 13h | 12h |
>
> > *2. My understanding is that the improvement offered by going from myopic to multi-step lookahead is not especially large. In some cases it helps noticeably, but in a number of problems the performance is almost the same. This would seem to reduce the significance of developing a multi-step method for this setting. Could the authors comment on whether the myopic variant of their algorithm is also new?*
>
> COPEx-myopic involves solving the constrained optimization problem (Section 3.4) at the root node without constructing the scenario tree. Since we did not find any existing work in the literature that solves the constrained BED problem, the myopic variant of our algorithm is also new.
>
> > *3. I'd like to understand how the algorithm behaves in a setting where I want to maximize EIG subject to a budget on the total cost of all observations. Given that the planning horizon H is typically short (1 to 3 steps ahead), is it correct that the algorithm would only take the budget into account when it can be completely consumed within H steps?...In light of the previous comment, I'm wondering what would happen if you changed the objective in the cost-aware active learning problem from the normalized EPIG to just the EPIG, and constrained the total cost. Would it still perform well relative to baselines?*
>
> Assuming a large budget relative to the planning horizon, we agree with your comment that the algorithm would only take budget into account towards the end when the budget is about to run out, and so the hard budget constraint would barely affect the decision. To verify this claim, we modified the experiment in Section 4.3 as per your suggestions: we placed a large budget compared to the planning horizon, and replaced the normalized EPIG with just EPIG. For comparison, we also tested a small budget case, where planning should help. The corresponding plots are in https://files.catbox.moe/lp45st.pdf. We see in Fig. 2(a) that when the budget is large compared to the planning horizon, COPEx-myopic performs equally well as COPEx-1 and COPEx-3, indicating that planning is not influencing the design choice as expected. However, when the budget is small (Fig. 2(b)), we observe performance gains compared to COPEx-myopic for Ackley and Goldstein Price tasks. This aligns with the results of the CES task in Table 1 where COPEx (H=1) and COPEx (H=3) perform better than COPEx-myopic, as the budget there can get exhausted in a few steps.
>
> Note that the situation you described has been addressed in the paper by Astudillo et al. (2021) using a “fantasy” budget. At each step, they compute the remaining budget, choose a cheap base policy to simulate $N$ future evaluations, add up the $N$ fantasy costs, and use that as the current budget. Thus, they replace the full remaining budget with the estimated cost of $N$ base-policy future steps, which then influences the planning at each step. Unfortunately, we were not able to empirically test this approach in the allotted rebuttal time, but we plan to see the effect of this strategy on our results.
>
>
> > *4. And is it correct that the one-step reward in the problem as implemented is the change in normalized EPIG?*
>
> Yes, the one-step reward is the change in normalized EPIG given in Eq. 9.
>
>
> > *5. The algorithm uses only short horizons (H=1,2,3) in the experiments. Several of the experiments also use a short horizon.*
>
> We have extended the location finding results up to H=5, see https://files.catbox.moe/u9gs33.pdf. Despite the cost scaling exponentially, the runtime is still within a couple of minutes for longer horizons due to the pre-trained inference and policy networks. However, we observe degradation in performance as H increases as the approximation errors accumulate, which further suggests that setting H to be large does not help.
>
>
> > *6. The paper has a nice limitations discussion in the conclusion. But if the answers to my other questions reveal additional limitations beyond what is discussed there, it should be added.*
>
> Good point. We will include the discussion of the large budget case in the limitations section.

---

### Official Review · Reviewer_dzk3 · 2026-03-13

**Soundness:** 3
**Presentation:** 3
**Significance:** 4
**Originality:** 3
**Overall Recommendation:** 4
**Confidence:** 4

**Summary:**

This paper introduces a semi-amortized Bayesian experimental design framework that fixes a critical gap in existing methods: standard amortized BED policies cannot adapt to dynamic, test-time constraints without full retraining. COPEx combines offline pre-trained amortized posterior and design policy networks with online multi-step lookahead planning via scenario trees, which explicitly enforces constraints across the planning horizon. The authors validate the method on three tasks: location finding with transition constraints, a CES behavioral economics task with global budget limits, and cost-aware active learning with spatially varying costs. Across all settings, COPEx consistently outperforms state-of-the-art BED and active learning baselines with only modest runtime overhead.
The authors examine the key problem of adapting principled BED to the path-dependent constraints ubiquitous in real-world sequential experimentation. The authors attempt to consider a relevant problem largely overlooked by the amortized BED literature, which typically assumes fixed, unconstrained design spaces during offline training and fails to generalize to unseen constraints at test time.

**Compliance With Llm Reviewing Policy:**

Affirmed.

**Key Questions For Authors:**

Can you provide any theoretical bounds, even asymptotic or probabilistic on COPEx’s approximation error relative to the optimal constrained BED policy?
Have you explored techniques to mitigate exponential runtime scaling with horizon H? Can you share results for H>3 and clarify the practical horizon limit for real-world use?

**Limitations:**

The authors include a dedicated Limitations section

**Strengths And Weaknesses:**

The paper correctly identifies that real-world sequential experiments are almost always subject to constraints that break standard amortized BED methods. Its unified constraint state formulation is flexible, capturing transition, budget, and heterogeneous cost constraints in a single framework without task-specific modifications. The authors find a smart balance between fully amortized (fast but inflexible) and full online planning (constraint-aware but computationally prohibitive) BED.
However this work provides no formal bounds on how close COPEx’s policy is to the optimal constrained BED policy, nor analysis of three core sources of approximation error: posterior bias, finite horizon truncation, and Monte Carlo EIG estimation. The authors note posterior error compounds over long horizons, but offer no theoretical analysis of this effect. and the scenario tree grows exponentially with planning horizon H, and the paper only tests H up to 3. No mitigation strategies are included for long-horizon tasks, nor analysis of the performance-runtime tradeoff for H>3.

---

> ### Author Rebuttal · Authors · 2026-03-30
>
> Thank you for your review and your positive assessment of our work. We address your questions below.
>
> > *1. Can you provide any theoretical bounds, even asymptotic or probabilistic on COPEx’s approximation error relative to the optimal constrained BED policy?*
>
> A full end-to-end optimality bound for COPEx is challenging, as the final policy error comes from four sources: finite-horizon truncation, posterior approximation, Monte Carlo EIG estimation, and non-convex tree optimization. We now try to make this decomposition explicit:
>
> Let $J_t^\star$ be the optimal constrained BED value and $J_t^{\text{COPEx}}$ denote the value induced by our $H$-step receding-horizon planner. Under bounded per-step utility $0\leq r_t \leq R_{\text{max}}$, we can write
> $$J_t^\star -  J_t^{\text{COPEx}} \leq \epsilon_{\text{trunc}}(H) + \epsilon_{\text{post}}(H) + \epsilon_{\text{MC}} + \epsilon_{\text{opt}},$$
> where $\epsilon_{\text{MC}} $ denotes the Monte Carlo estimation error and $\epsilon_{\text{opt}} $ denotes optimization error from approximately solving the non-convex tree objective.
>
> Truncation error $\epsilon_{\text{trunc}}(H) $: As we use an $H$-step receding-horizon approximation, the lost tail value is bounded by
> $\epsilon_{\text{trunc}}(H) \leq \sum_{\ell = H+1}^{T-t} \gamma^\ell R_{\text{max}} $, which is at most $(T-t-H)R_{\text{max}} $ when $\gamma=1$.
>
> Posterior/predictive error $\epsilon_{\text{post}}(H)$: Training $q_{\hat \phi}$ by maximum likelihood corresponds, at the population level, to minimizing conditional cross-entropy, whose excess risk equals an expected KL divergence to the true posterior up to an additive constant. Hence, with sufficient model capacity, representative simulated training data, and successful optimization, $q_{\hat \phi}$ can approach the true posterior as the amount of training data increases.
>
> In planning, if the one-step posterior-predictive approximation error induced by $q_{\hat \phi}$ is uniformly bounded by $\epsilon_\phi$, then the induced error in the $H$-step lookahead value accumulates across the rollout and scales at most linearly with the effective horizon:
> $\epsilon_{\text{post}}(H)  = O(\sum_{\ell=0}^{H-1} \gamma^\ell \epsilon_\phi)$,
> i.e., $O(H \epsilon_\phi)$ when $\gamma=1$. This formalizes our empirical observation that deeper lookahead can degrade when the amortized posterior is imperfect.
>
> For EIG estimation, Eq. (7) reduces to the ACE lower bound of Foster et al. (2020) when $q_\hat{\phi} = p(\theta|D)$. In that exact-posterior case, the bound tightens monotonically with the number of contrastive samples $L$. When $q_\hat{\phi} \neq p(\theta|D)$, Eq. (7) should be interpreted as a surrogate utility, and its deviation from the true EIG can be decomposed into (i) posterior approximation bias and (ii) Monte Carlo error. Under finite-variance assumptions, the Monte Carlo component $\epsilon_{\text{MC}}$  decreases at the usual $\frac{1}{\sqrt{L}}$ rate, while scenario-tree averaging over $m$ fantasy branches contributes an additional $\frac{1}{\sqrt{m}}$ dependence.
>
> Thus, while we do not claim a single tight global optimality theorem in the current paper, the main approximation terms can be bounded under standard assumptions. We will add this decomposition and discussion in the revised version.
>
> > *2. Can you share results for H>3 and clarify the practical horizon limit for real-world use?*
>
> Absolutely. The results for H>3 in the location finding task can be seen here https://files.catbox.moe/u9gs33.pdf. We observe that the performance begins to degrade beyond H=3, especially for the most constrained case of $\delta = 0.05$. In practice, we expect the ideal horizon limit to be a function of the specific example and constraint. However, given our findings in different cases, we recommend a limit of H=2 or H=3, which would already be an improvement upon the baseline BED methods, while keeping the online cost minimal.
>
> > *3. Have you explored techniques to mitigate exponential runtime scaling with horizon H?*
>
> Reducing the computational cost of simulating longer horizon scenario trees is indeed useful for applications requiring deeper lookahead. Existing techniques such as DESPOT (Somani et al., 2013) and Hyp-DESPOT (Cai et al., 2021), which employ particle-based belief approximations and branch pruning, have proven effective in the context of POMDPs and could in principle be adapted to the BED setting. We did not explore such techniques as the cost of generating the tree was in the order of a couple of minutes at max, even with the exponential scaling, primarily due to the pre-trained inference and policy networks. For instance, COPEx with H=1 took around 19s to generate the tree, while it took around 2-3 mins for H=5, which is not computationally prohibitive. Moreover, as performance degrades with a longer horizon, adapting such techniques is not very beneficial in our view.

---

### Official Review · Reviewer_Bss8 · 2026-03-13

**Soundness:** 2
**Presentation:** 3
**Significance:** 3
**Originality:** 3
**Overall Recommendation:** 4
**Confidence:** 1

**Summary:**

This paper presents a Bayesian experimental design framework that utilizes an amortized posterior network with a scenario tree and a multi-step lookahead in an online fashion. The design policy is trained offline to assist in accelerated state updates. The pretrained design policy also provides a high-quality initial design.

**Compliance With Llm Reviewing Policy:**

Affirmed.

**Final Justification:**

My questions were answered properly and I increased the score for presentation and significance accordingly. The explanation of unconstrained training and constrained, budgeted deployment was helpful. It would have been easier to assess the innovation if it had been demonstrated on extrapolation benchmarks, A/B testing, material design tasks, or economic preference benchmarks for Bayesian Experimental Design (BED). Since I am unable to compare the proposed method to the state of the art in BED, I leave this comment as an educated guess.

**Key Questions For Authors:**

My main question is how the Branin and Ackley functions here are not simply mathematical optimization tasks, and what makes them fall into Bayesian Experimental Design (BED) benchmarks?

**Limitations:**

Yes

**Strengths And Weaknesses:**

Strengths:

Figure 2 lays out the framework adequately for reader comprehension.


Weaknesses:

I was not able to follow any mathematical path for why this framework would converge to an optimal policy to maximize the EIG reward, or why the suggested method works better.

The experiments to me (such as Branin and Ackley functions) are mathematical optimization rather than Bayesian Experimental Design benchmarks. I would expect extrapolation benchmarks and A/B testing, material design, or economic preference benchmarks for Bayesian Experimental Design.

---

> ### Author Rebuttal · Authors · 2026-03-30
>
> Thank you for your review. We address your comments/questions below:
>
> > *1. I was not able to follow any mathematical path for why this framework would converge to an optimal policy to maximize the EIG reward, or why the suggested method works better.*
>
> Consider the case where a design policy is trained offline in an unconstrained setting, but constraints related to budget or design are enforced at deployment. In such cases, the learned policy will be sub-optimal. The idea of this method is to adapt that pre-trained policy online by solving a constrained optimization task (i.e., maximize EIG under the given constraints, see Eq. 2). We solve this by framing it as a (finite-horizon) planning problem. This method works better because it explicitly takes into account the constraints when choosing the next design, while existing BED methods do not have that capability. Hope that makes it clearer.
>
> > *2. My main question is how the Branin and Ackley functions here are not simply mathematical optimization tasks, and what makes them fall into Bayesian Experimental Design (BED) benchmarks?*
>
> Indeed the Branin and Ackley are mathematical functions and not probabilistic models, which would be required for BED benchmarks. However, we are running an active learning experiment instead of an optimization task in Section 4.3. That is, we sequentially collect data that helps us learn Branin and Ackley functions in terms of reducing the predictive error/entropy, measured via the expected predictive information gain (EPIG). So the task is not to optimize the function, but to infer the predictive distribution, which makes it fall under a prediction-oriented BED task [1] (instead of the usual posterior inference BED task). This experiment shows the flexibility of our method as it is applicable to learning about both the parameter and the predictive distribution.
>
> [1] Smith et al. "Prediction-oriented Bayesian active learning." AISTATS 2023.

---

> > ### Author Rebuttal · Reviewer_Bss8 · 2026-04-04
> >
> > My questions were answered properly and I increased the score for presentation and significance accordingly. The explanation of unconstrained training and constrained, budgeted deployment was helpful. It would have been easier to assess the innovation if it had been demonstrated on extrapolation benchmarks, A/B testing, material design tasks, or economic preference benchmarks for Bayesian Experimental Design (BED). Since I am unable to compare the proposed method to the state of the art in BED, I leave this comment as an educated guess.

---

> > > ### Author Response · Authors · 2026-04-06
> > >
> > > We sincerely thank the reviewer for engaging with our rebuttal and for acknowledging that the main concerns were resolved.
> > >
> > > To clarify the positioning of our empirical evaluation, our paper focuses on Bayesian experimental design, where the goal is to reduce uncertainty about unknown model parameters through informative experiments (Rainforth et al. 2024). Our experiments already include established benchmarks from the modern BED literature. In particular, the CES task is precisely an *economic preference* elicitation problem: it models a participant's utility over baskets of goods and aims to infer latent preference parameters. We also study the Location Finding task, which is a standard benchmark used in several prior BED works (Foster et al., 2019; 2021; Ivanova et al., 2021; Blau et al., 2022; Huang et al., 2025). Regarding the other suggested application domains, materials design is often associated with Bayesian optimization, while A/B testing is more commonly discussed in the literature on traditional experimentation and adaptive bandits. We are not aware of BED papers with those benchmarks. If the reviewer has references in mind, we would be genuinely grateful if they could point us to them.
> > >
> > > We also note that our comparisons do include strong *state-of-the-art* BED baselines. In particular, ALINE (Huang et al., 2025) is a recent leading amortized BED method, and VPCE (Foster et al., 2020) is a well-established non-amortized baseline. COPEx consistently outperforms both in our experiments. If the reviewer had other state-of-the-art BED methods in mind, we would also appreciate those references.

---

### Official Review · Reviewer_4pRs · 2026-03-13

**Soundness:** 2
**Presentation:** 3
**Significance:** 3
**Originality:** 3
**Overall Recommendation:** 4
**Confidence:** 4

**Summary:**

The paper studies Bayesian experimental design under deployment-time constraints, where the feasible action set may depend on previous designs, remaining budget, or other evolving state variables.

The authors propose COPEx, a semi-amortized method that combines offline-trained amortized models (posterior/predictive network and proposal policy) with online scenario-tree planning. The amortized posterior enables cheap belief updates and fantasy observations during planning, while the proposal policy initializes constrained optimization over the design variables. The approach enables non-myopic sequential design under constraints without retraining.

Empirical evaluation on constrained location finding, CES preference elicitation with global budget constraints, and cost-aware active learning shows improvements over myopic and unconstrained baselines, particularly with short-horizon planning.

**Compliance With Llm Reviewing Policy:**

Affirmed.

**Key Questions For Authors:**

1. The planner relies on amortized posterior predictions and approximate EIG/EPIG estimates. Can the authors quantify the accuracy of these surrogates (e.g., on smaller problems where exact posterior updates are feasible) to better understand when planning errors accumulate?

2. In the active learning experiments, to what extent do improvements come from non-myopic planning vs. continuous constrained optimization of design variables?

3. Performance appears sensitive to planning depth (e.g., degradation at larger horizons in CES). Can the authors provide guidance on the practical regime for horizon and branching factor where the method remains reliable?

4. The framework assumes amortized models trained under a distribution of tasks/constraints. How robust is the method when deployment constraints differ significantly from those seen during amortized training?

**Limitations:**

Yes

**Strengths And Weaknesses:**

**Strengths**

* The paper addresses an important and realistic gap in Bayesian experimental design: test-time constraints that depend on sequential decisions (e.g., budget or transition constraints).

* The formulation as planning in an augmented state space is technically sound and integrates naturally with amortized BED components.

* Amortized posterior models provide tractable belief updates while scenario-tree planning captures non-myopic effects.

* Empirical evaluation spans multiple tasks and includes useful ablations (horizon length, branching factor, initialization, runtime).

**Weakness**

* The online planner optimizes a surrogate objective built from amortized posterior predictions and approximate EIG/EPIG estimators. The paper does not directly quantify how approximation error affects planning quality, especially for longer horizons.

* Some empirical comparisons are difficult to attribute precisely. In the active learning experiments, part of the gain may come from continuous constrained optimization of design variables rather than purely from non-myopic planning, since several GP baselines operate on discrete candidate pools.

* Empirical validation remains limited to relatively controlled benchmarks; it is unclear how robust the approach is, e.g., in higher-dimensional design spaces or under stronger distribution shift between training and deployment constraints.

---

> ### Author Rebuttal · Authors · 2026-03-30
>
> Thank you for your positive assessment of our work and the points you raised.
>
> > *1. Can the authors quantify the accuracy of these surrogates to better understand when planning errors accumulate?*
>
> Since both fantasy outcome generation and EIG estimation in our planning framework rely on the amortized posterior​​, its accuracy drives scenario-tree quality. Consistent with the error decomposition in our response to Reviewer dzk3, we empirically quantify how posterior approximation error accumulates with horizon. On the location finding task, we can use importance sampling (IS) to obtain a high-fidelity reference posterior. We run two fantasy rollout pipelines using identical designs at each depth: one using our amortized posterior, and one using an IS-based posterior. At each depth $H$, we measure the Maximum Mean Discrepancy (MMD) between the posterior samples from the two pipelines, averaged over 100 independent rollouts:
>
> | Metrics (±SE) | H=0 | H=1 | H=2 | H=3 | H=4 | H=5 |
> | :---- | :---- | :---- | :---- | :---- | :---- | :---- |
> | MMD   | 0.42 ± 0.004 | 0.48 ± 0.004 | 0.53 ± 0.004 | 0.58 ± 0.003 | 0.60 ± 0.004 | 0.64 ± 0.004 |
>
> We observe that posterior divergence increases monotonically with depth, confirming that approximation errors from the amortized surrogate compound along fantasy rollouts. This supports the intuition in the paper that deeper lookahead becomes more sensitive to surrogate error.
>
> > *2. In the active learning experiments, to what extent do improvements come from non-myopic planning vs. continuous constrained optimization of design variables?*
>
> You are right that part of the gains in the active learning task come from the GP baselines operating on discrete space. We have now made all the GP baselines continuous for a fair comparison, see https://files.catbox.moe/8nzwzy.pdf. The performance of GP-US and GP-VR has now improved. COPEx still outperforms the baselines for Ackley and Goldstein Price functions, while in the case of Branin, the GP-US baseline performs equally well. We will replace the existing Figure 4 from the paper with this result and update the discussion accordingly.
>
> Inspired by Reviewer 1v3e’s comments, we conducted a modified AL experiment with a hard budget constraint where we use EPIG instead of the cost-normalized EPIG. The performance gap between the baselines and different variants of COPEx becomes larger in this case, see https://files.catbox.moe/ajiq1h.pdf.
>
> > *3. Can the authors provide guidance on the practical regime for horizon and branching factor where the method remains reliable?*
>
> We have extended the location finding results up to H=5, see https://files.catbox.moe/u9gs33.pdf. We observe that the performance begins to degrade beyond H=3, especially for the most constrained case of $\delta = 0.05$. In practice, we expect the ideal horizon limit to be a function of the specific example and constraint. However, given our findings in different cases, we recommend a limit of H=2 or H=3, which would already be an improvement upon the baseline BED methods, while keeping the online cost minimal.
>
> > *4. How robust is the method when deployment constraints differ significantly from those seen during amortized training?*
>
> Thank you for the opportunity to clarify this. The amortized posterior network and policy in COPEx are both not trained on any specific constraint. The posterior network is trained purely on random design-observation pairs sampled from the generative model independent of any constraint. Similarly, the amortized policy is trained on the unconstrained BED objective. Constraints are handled entirely at test time through the online planning and optimization procedure, where the constrained optimizer enforces feasibility over the scenario tree.
>
> Nevertheless, your question raises a valid concern: when constraints are very tight, the amortized policy may propose initializations that lie far from the constrained optimum, since it was trained without constraints. We address this through a hybrid exploration–exploitation initialization scheme, and we empirically observe its benefit in the active learning experiments.
>
> With regards to training the amortized network under a distribution of tasks, this indeed relies on the assumption that the likelihood model we are using to generate simulations (i.e. training data) is well-specified. Thus, if the model is misspecified, meaning that the true data-generating process does not belong in the model family, amortized inference methods can get heavily biased. This is still an active area of research in simulation-based inference [1], and hence can impact our method as well. In such cases, we can use some of the robust methods e.g. Huang et al. (2023) to reduce the impact of misspecification. We will emphasise this point further in the limitations section.
>
> [1] Kelly et al. "Simulation-based Bayesian inference under model misspecification." arXiv:2503.12315.

---

> > ### Author Rebuttal · Reviewer_4pRs · 2026-04-03
> >
> > Thanks for the detailed responses, which addresses my main concerns.
> >
> > The surrogate error (MMD vs. depth) analysis is consistent with the observed degradation. The updated active learning results with continuous baselines also help better isolate the effect of planning. The guidance on practical horizon and clarification on handling constraints at test time are helpful.
> >
> > Overall, concerns are largely resolved. I’m comfortable maintaining a weak accept.

---

> > > ### Author Response · Authors · 2026-04-06
> > >
> > > We sincerely thank the reviewer for taking the time to read our rebuttal in detail. We greatly appreciate your acknowledgement that the main concerns have been addressed.

---

### Decision · Program_Chairs · 2026-04-30

**Decision:**

Accept (regular)

**Comment:**

The paper introduces COPEx, a semi-amortized framework for Bayesian experimental design (BED) under dynamic constraints, combining offline pre-trained amortized posterior and policy networks with online scenario-tree planning. All four reviewers agree the paper addresses an important and underexplored gap in BED (handling test-time constraints such as budgets, varying costs, and transition restrictions) and that the proposed approach is principled and well-motivated. The paper is clearly written, and the empirical evaluation spans multiple tasks with useful ablations.

The main limitations are the lack of formal optimality guarantees and the exponential scaling of the scenario tree with planning horizon, which restricts practical use to short horizons (H=2 or 3). The authors provided a helpful error decomposition in the rebuttal and extended experiments to H=5, showing that performance degrades beyond H=3 due to compounding approximation errors. Reviewer 1v3e also raised a thoughtful point about budget constraints only becoming active near the end of the horizon, which the authors verified empirically and acknowledged as a limitation.

None of the reviewers were willing to champion the paper, so while the sentiments are largely positive, I can only recommend this paper for a weak accept. In the revision (either the final version or next submission), I encourage the authors to include the experiments iwth H=5 in the main text and clearly discuss the limitations.